

# Spatial-temporal patterns of anthropogenic and biomass burning contributions on air pollution and mortality burden changes in India from 1995 to 2014

Bin Luo[1], Yuqiang Zhang[1], Tao Tang[2,3], Hongliang Zhang[4], Jianlin Hu[5], Jiangshan Mu[1], Wenxing Wang[1], Likun Xue[1]

[1]Environment Research Institute, Shandong University, Qingdao 266237, China
[2]School of the Environment, Yale University, New Haven, CT 06511, USA
[3]Institute of Atmospheric Physics, Chinese Academy of Sciences, Beijing 100029, China
[4]Department of Environmental Science and Engineering, Fudan University, Shanghai 200438, China
[5]Jiangsu Key Laboratory of Atmospheric Environment Monitoring and Pollution Control, Collaborative Innovation Center of Atmospheric Environment and Equipment Technology, Nanjing University of Information Science & Technology, Nanjing 210044, China

*Correspondence to*: Yuqiang Zhang (yuqiang.zhang@sdu.edu.cn), Likun Xue (xuelikun@sdu.edu.cn)

**Abstract.** Anthropogenic and biomass burning emissions are the major sources of ambient air pollution. India has experienced a dramatic deterioration in air quality over the past few decades, but no systematic assessment has been made to investigate the individual contributions of anthropogenic and biomass burning emissions. In this study, we conducted a pioneering comprehensive analysis of the long-term trends of particulate matter with aerodynamic diameters < 2.5 μm ($PM_{2.5}$) and ozone ($O_3$) in India and their mortality burden changes from 1995 to 2014, using a state-of-the-art high-resolution global chemical transport model (CAM-chem). Our simulations revealed a substantial nationwide increase in annual mean $PM_{2.5}$ (6.71 μg m$^{-3}$ decade$^{-1}$) and $O_3$ (7.08 ppbv decade$^{-1}$), with the Indo-Gangetic Plain (IGP) and eastern central India as hotspots for $PM_{2.5}$ and $O_3$ trend changes individually. Noteworthy substantial $O_3$ decreases were observed in the northern IGP which were potentially linked to NO titration due to a surge in NOx emissions. Sensitivity analyses highlighted anthropogenic emissions as primary contributors to rising $PM_{2.5}$ and $O_3$, while biomass burning played a prominent role in winter and spring. In years with high biomass burning activity, the contributions from BB on both $PM_{2.5}$ and $O_3$ changes were comparable with or even exceeding anthropogenic emissions in specific areas. The elevated air pollutants were associated with increased premature mortality attributable to $PM_{2.5}$ and $O_3$, leading to 97.83 K and 73.91 K per decade. Despite a per capita decrease in the IGP region, the increased population offset its effectiveness.

## 1 Introduction

Air pollution is among the most detrimental environmental factors to human health. According to the World Health Organization (WHO) database, 99 % of the global population lives in areas where air quality surpasses WHO guideline limits (WHO database). The two most concerned pollutants, particulate matter with aerodynamic diameters < 2.5 μm ($PM_{2.5}$)



and ozone ($O_3$), can cause significant damage to the human heart and lungs (Hoek et al., 2013; Hystad et al., 2013; Villeneuve et al., 2015), potentially leading to premature death when exposed over extended periods (Dedoussi et al., 2020; Fuller et al., 2022). The latest Global Burden Disease (GBD2019) study estimated that exposure to air pollution, including

both household and ambient pollution, led to 6.7 million premature deaths (95 % confidence interval [CI], 5.9 to 7.5 million) worldwide in 2019 (Murray et al., 2020). Thus, the urgency of dealing with air pollution has become one of the most pressing global challenges.

It is well-known that surface air pollution is usually unequally distributed in space, with higher levels in developing countries than in developed countries (Forouzanfar et al., 2016). For example, India was ranked as the most polluted country

in the world in 2021, with 63 of the world's 100 most polluted cities (IQAir, 2022). Previous modeling studies indicated that districts exceeding India's annual ambient standard of 40 μg m$^{-3}$ rose from 200 to 385 out of 640 from 1998 to 2020 (Guttikunda and Ka, 2022). The GBD2019 study estimated that premature deaths attributed to ambient $PM_{2.5}$ and $O_3$ pollution accounted for 10.4 % (8.4-12.3) and 1.8 % (0.9-2.7) of the total deaths in India in 2019, respectively, and the death rate per 100,000 people increased by 115.3 % (28.3-344.4) and 139.2 % (96.5-195.8) from 1990 to 2019, respectively

(Pandey et al., 2021). Meanwhile, the faster chemical reaction rates in India due to the strong convection, sunlight, and warm temperatures, making it a hot spot for accumulating major air pollutants compared with other regions and easily affecting the air quality in downwind regions (Zhang et al., 2016, 2021a).

As seen from the Community Emissions Data System (CEDS) inventory (Hoesly et al., 2018), the increasing trends of anthropogenic (ANTHRO) emissions of major air pollutants, such as nitrogen oxides (NOx), carbon monoxide (CO), and

non-methane volatile organic compound (NMVOC), are significantly higher in India than those in other regions (Wang et al., 2022). Meanwhile, crop yields in India have significantly enhanced since the mid-1960s after the Green Revolution, contributing to increased biomass burning (BB) emissions (Huang et al., 2022). The study showed that from 1950–51 to 2017–18, the crop residue burning in India increased from 18 million tonnes to 116 million tonnes in terms of total biomass burned (Venkatramanan et al., 2021). The frequency and intensity of forest fires in India have also increased in recent years

due to persistent warmer temperatures and climate extremes (Vadrevu et al., 2019; Jain et al., 2021). These in turn could pose significant threats to ambient air quality and human health because large amounts of compounds are emitted into the atmosphere, namely carbon dioxide ($CO_2$), NOx, particulate matter (PM), and other chemical species (Crutzen and Andreae, 1990; Carvalho et al., 2011; Lan et al., 2022; Miranda et al., 2005).

In this study, we aim to improve our understanding of the spatial-temporal distribution of major air pollutants, mainly

surface $PM_{2.5}$ and $O_3$, and the related mortality burden in India from 1995-2014 using a state-of-the-art global chemistry transport model. In addition, the individual contributions of changes in ANTHRO and BB emissions were further separated to better understand the causes of worsening air quality and escalating health risks in India. The selected period encompasses a dynamic phase of rapid changes in both anthropogenic and biomass burning activities in India, thereby providing an ideal context for investigating their respective contributions to air pollution.



## 2 Methods

### 2.1 CAM-chem model configuration


We simulated surface PM$_{2.5}$ and O$_3$ concentrations over India between 1995 and 2014 using the global chemistry model CAM-chem, which is based on version 6 of the Community Atmosphere Model (CAM6), the atmospheric component of the Community Earth System Model (CESM2), as detailed by Danabasoglu et al. (2020) and Emmons et al. (2020). Following

Emmons et al. (2020) the original model was run at 1.25° (longitude) × 0.9° (latitude) horizontal resolution with 32 vertical levels reaching ~45 km. We configured the Model of Ozone and Related Chemical Tracers Tropospheric and Stratospheric (MOZART-TS1) chemistry mechanism with various complexity choices for tropospheric and stratospheric chemistry (Emmons et al., 2020). The aerosol module adopted the four-mode version of the Modal Aerosol Model (MAM4), including sulfate, black carbon, primary organic matter, secondary organic aerosols, sea salt, and mineral dust. The first level of the

model outputs was considered the surface level, and all the model outputs were then regridded to a finer resolution 0.5° × 0.5° to match the grid-cell population and baseline mortality rates datasets in performing the health impact assessment.

Global historical ANTHRO emissions were adopted from CEDS (version 2017-05-18), which provides monthly emissions of anthropogenic aerosol and precursor compounds at 0.5° × 0.5° from 1750 to 2014 and were used in the Coupled Model Intercomparison Project Phase 6 (CMIP6) experiments (Emmons et al., 2020; Hoesly et al., 2018). The air pollutants

from the CEDS inventory, especially the NMVOC, were then re-speciated to match the chemical species in the latest CESM2 model, following the steps introduced by Emmons et al. (2020). BB emissions were sourced from van Marle et al. (2017) at 0.5° native resolution and were all emitted at the surface.

### 2.2 Numerical experiments designs

The standard simulation (BASE) was driven by year-varying ANTHRO and BB emissions from 1995 to 2014, as

described above. To separate the contributions from these two emission sources, we then conducted two sensitivity simulations in which ANTHRO emissions (FixAN) and BB emissions (FixBB) were fixed at 1995 levels individually, while all other parameters were kept consistent with the BASE (Table 1). Subtracting the BASE from each sensitivity enables quantifying the influences of changes in ANTHRO and BB emissions on air quality and its associated health burden in India, respectively.

**Table 1. Model simulations performed in this study**

| Simulation | Anthropogenic emissions | Biomass burning emissions |
|:---:|:---:|:---:|
| BASE | V | V |
| FixAN | 1995 | V |
| FixBB | V | 1995 |



"V" indicates that particular input is subject to interannual variation in the simulation during the period 1995-2014, "FixAN" indicates that only global ANTHRO emissions were set to 1995 conditions in the simulation. "FixBB", indicates that only global BB emissions were set to 1995 level.

### 2.3 Trend estimation

In this study, we applied the Theil-Sen estimator (Theil, 1992; Sen, 1968) to calculate the magnitude of trends in surface PM$_{2.5}$ and O$_3$ concentrations and the attributed mortality burden spanning from 1995 to 2014. The Theil-Sen estimator is a robust non-parametric method for trend analysis based on the median slope, which is insensitive to outliers and highly competent in identifying the slope of non-normally distributed data, as described in eq 1. This method has been widely used to analyse temporal trends in air pollutants that are always non-normally distributed (e.g., Munir et al., 2013;

Sarkar et al., 2019; Vanem and Walker, 2013; Wan et al., 2023).

$$Slope = Median \frac{(x_i - x_j)}{(t_i - t_j)} \tag{1}$$

Where $x_i$ and $x_j$ represent the PM$_{2.5}$ and O$_3$ concentrations or attributed premature mortality at time $t_i$ and $t_j$ $(i > j)$, respectively. $Slope > 0$ indicates an increasing trend; $Slope < 0$ indicates a decreasing trend.

In complement to the Theil-Sen estimator, we used the nonparametric Mann-Kendall test to assess the significance of temporal trends within the data series (Zhang et al., 2022a, b). According to previous studies, p-value less than 0.05 is most

commonly treated as the absolute threshold of statistical significance (Christiansen et al., 2020; Wang et al., 2021; Zhou et al., 2017). The above methods were completed by implementing a Python program with the package "pymannkendall", as detailed at https://pypi.org/project/pymannkendall/, last accessed on March 20, 2024. We will discuss the air quality and mortality burden changes in six Indian regions based on meteorological conditions and aerosol variability (Fig. S1).

### 2.4 Mortality burdens of surface PM$_{2.5}$ and O$_3$ in India

Based on an integrated exposure-response function utilized in the most recent GBD studies, we estimated the mortality burden associated with long-term exposure to ambient annual PM$_{2.5}$ and 6-month running average of daily maximum 8-hr average (MDA 8) O$_3$ in India spanning from 1995 to 2014, as described in eq 2.

$$\Delta Mort = y_0 \times AF \times pop = y_0 \times \left(\frac{RR-1}{RR}\right) \times pop \tag{2}$$

Where $\Delta Mort$ refers to the annual mortality burden attributed to long-term PM$_{2.5}$ or O$_3$ exposure, and $y_0$ is the baseline mortality rate for a specific cause of disease. $AF$ is the attributable fraction measuring the PM$_{2.5}$ or O$_3$ exposure attributable

disease burden, which is represented by $\frac{RR-1}{RR}$ ($RR$ refers to relative risk). $pop$ represents the exposed population above the age of 25 for each grid cell in the domain.





Following our previous work (Zhang et al., 2021b), we obtained the baseline mortality rate ($y_0$) for each country and 5-year age group from 1995-2014 from the GBD2017 project (Stanaway et al., 2018). The $RR$ of long-term PM$_{2.5}$ exposure associated with mortality burden due to specific disease was estimated using an integrated exposure-response model (IER) constructed by Burnett et al. (2014) and updated in GBD2017. The $RR$ for long-term O$_3$ exposure was obtained from Turner et al. (2016) which indicated an $RR$ of 1.12 (95 % confidence interval (CI): 1.08, 1.16) for respiratory disease. The recent GBD2019 reported a relatively lower $RR$ for the chronic obstructive pulmonary disease (COPD), a subcategory of respiratory disease (1.06, with 95 % CI: 1.03, 1.10). To be comparable with the GBD2019 results, we also estimated the O$_3$-related mortality burden for the COPD in India during the same period. Population distribution with age stratification data ($pop$) was retrieved from the GBD2017. The population-weighted average of specific air pollutants discussed in the results was calculated by weighting the population of all grid cells inside each administrative region or country. Additionally, we calculated mortality rates per capita (avoid deaths per 100,000 people) in each administrative region to exclude the influence of varying populations.

## 3 Results and discussion

### 3.1 CAM-chem evaluation

We performed a comprehensive model evaluation by comparing our simulated monthly concentrations from the BASE with multiple datasets, including ground-based observations in India, historical multi-model simulation from the CMIP6 project, and different versions of multi-year reanalysis data from the Atmospheric Composition Analysis Group (ACAG) at Washington University in St. Louis, hereinafter referred as 'Wustl Extracts' (van Donkelaar et al., 2021). We also compared our simulated PM$_{2.5}$ and O$_3$ with previously published studies in India using either global or regional chemical transport models (CTMs), as well as the concentration reported from the GBD2019. We selected available ground-level PM$_{2.5}$ observations over India from previous studies (Latha and Badarinath, 2005; Panwar et al., 2013; Reddy et al., 2012; Saradhi et al., 2008; Tiwari et al., 2009, 2013), which were also collected by the ACAG. Figure S2 indicates that the model exhibits good performance in capturing seasonal variations of surface PM$_{2.5}$ observations, especially during the peak months, with correlation coefficients (R) ranging from 0.59 to 0.91. Two exceptions are Mumbai (with R of -0.16), where the model shows a contrasting trend for the seasonal PM$_{2.5}$ characteristics (Fig. S2b), and Mukteshwar (with R of 0.45). One possible explanation is the potential underestimation of emission inventories, especially during early periods for developing regions, such as India (McDuffie et al., 2020; Wang et al., 2022; Agarwal et al., 2024). For O$_3$, our model shows an even higher R when compared with the available surface observation sites in India from 1997 to 2011 (Fig. S3). Unlike the underestimations of surface PM$_{2.5}$ in India, the CAM-chem model tends to overestimate surface O$_3$, which was not very uncommon for global CTM and also frequently discussed in previous studies (Hou et al., 2023; Tilmes et al., 2015; Young et al., 2018; Zhang et al., 2021b). The overestimation was partly caused by the coarse resolution, which leads to diluted





emissions of $O_3$ precursors and then simulated high $O_3$ production. Figure 1 compared our study with several previous studies and other publicly available $PM_{2.5}$ and $O_3$ datasets, as detailed in Tables S1 and S2. The comparisons indicate our

simulated results using the CAM-chem agree very well with previous studies for both $PM_{2.5}$ and $O_3$, based on either the various metrics (such as annual average and 6-month MDA8 $O_3$) or the population-weighted averages, consistent with the findings within the multiple CMIP6 models (Turnock et al., 2020). Figure S4 further compares the long-term trend of annual surface $PM_{2.5}$ concentrations from 1998 to 2014 in the BASE and Wustl Extracts dataset. A consistent increasing trend was found in both datasets, with temporal R of 0.86 and lower estimations in our model. The model performs better in eastern

India than in western India, with R usually being larger than 0.9 and NMB lower than -25 %. Similarly, compared to the simulated trend in our study with different versions of Wustl Extracts and the GBD2019, our simulated $PM_{2.5}$ concentration is lower, and the simulated $O_3$ is higher (Fig. S5). The underestimation of the surface $PM_{2.5}$ was partly caused by the missing model representation of nitrate and ammonium (Ren et al., 2023) and the secondary organic aerosol (Liu et al., 2021).

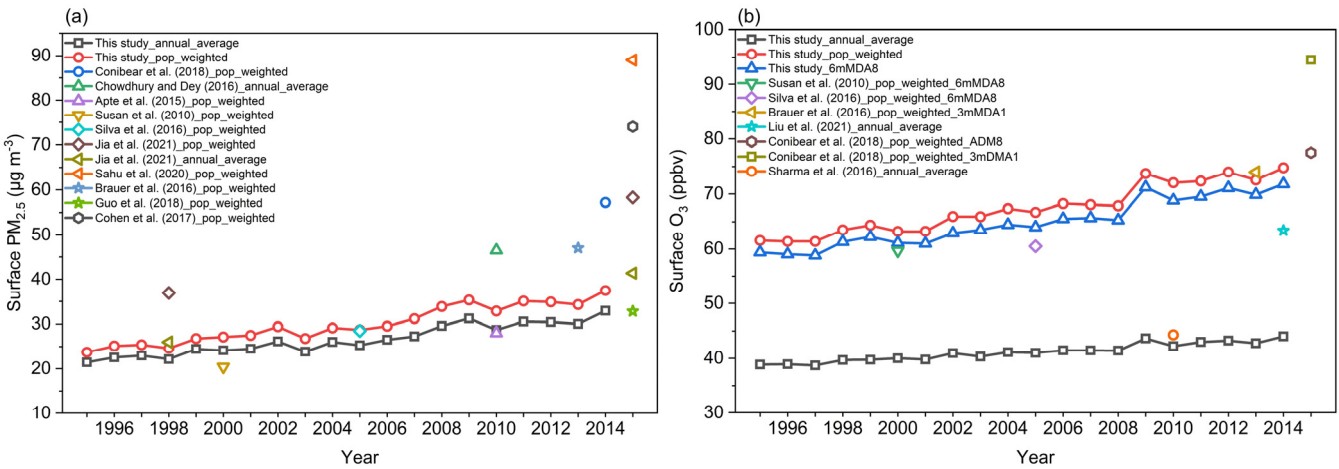

**Figure 1. Comparison of annual $PM_{2.5}$ and $O_3$ concentrations in India with previous studies. Note that the metrics vary depending on the study.**

## 3.2 Spatial and temporal distribution of air pollution changes in India from 1995 to 2014

### 3.2.1 Historical emissions in India from 1995 to 2014

We first assessed the interannual variation of ANTHRO and BB emissions of CO, NOx, NMVOC, sulfur dioxide ($SO_2$),

ammonia ($NH_3$), black carbon (BC), and organic carbon (OC) in India between 1995 and 2014 from the CEDS. Figure S6 indicates an overall increase in ANTHRO emissions before slowly falling after 2011. Significant inter-annual variations for BB emissions, such as in 1999, 2006, and 2009, were mainly caused by climate change-induced hot and arid conditions (Sahu et al., 2015). Figure S7 shows that ANTHRO emissions occurred predominately in IGP and central India, significantly increasing across all regions. Unlike other administrative regions, northern and eastern India, such as Punjab and Manipur,

features higher BB emissions and lower ANTHRO emissions.



### 3.2.2 The long-term trends of PM$_{2.5}$ and O$_3$ in India from 1995 to 2014

From the BASE simulation, we estimated that the annual mean population-weighted PM$_{2.5}$ and O$_3$ in India were 29.88 µg m$^{-3}$ and 67.41 ppbv from 1995 to 2014. Figure 2a, b shows that PM$_{2.5}$ concentrations gradually rise from the south to the north, with high levels predominantly found in the IGP, mainly caused by high ANTHRO emissions (Fig. S7) and reduced
ventilation due to obstruction by the Tibetan Plateau (Gao et al., 2018). Unlike PM$_{2.5}$, surface O$_3$ concentrations gradually increase from west to east and south to north, with the highest levels concentrated in northern India. The spatial patterns of the PM$_{2.5}$ and O$_3$ distribution in India were also seen in several previous studies, though they only discussed one or several specific years (Jia et al., 2021; Pandey et al., 2021).

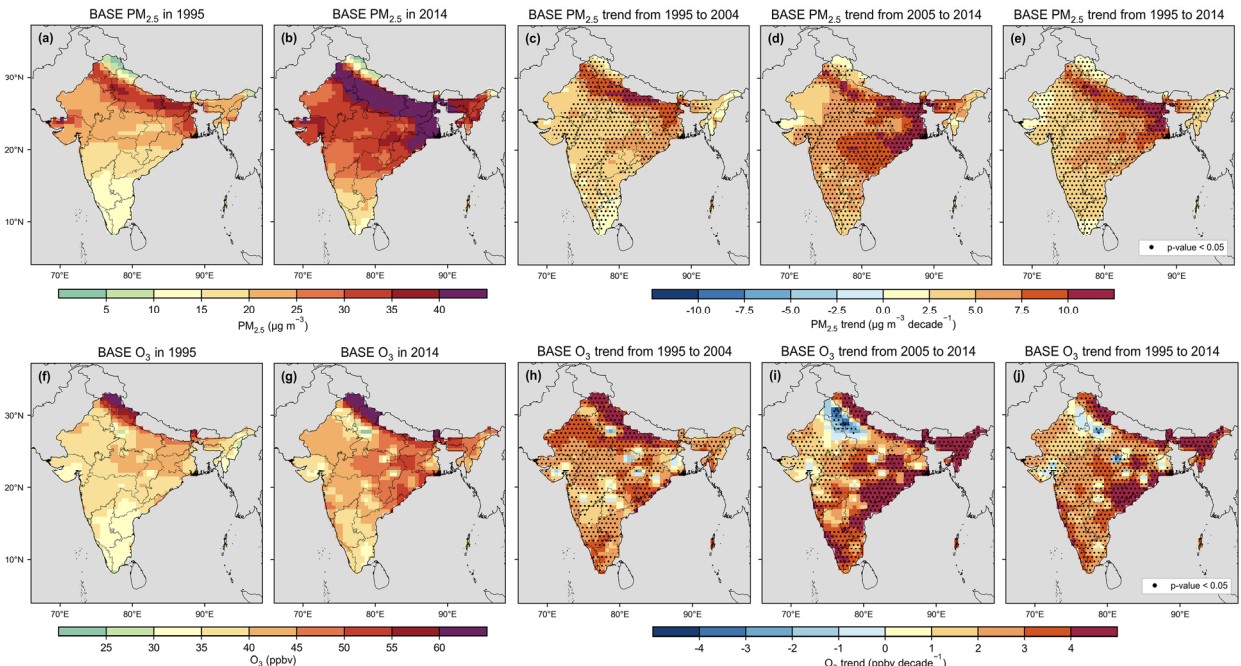

**Figure 2. Spatial distributions of PM$_{2.5}$ (top panel) and O$_3$ (bottom panel) for annual average in 1995 (a, f) and 2014 (b, g), with the trends from 1995 to 2004 (c, h), 2005 to 2014 (d, i), and 1995 to 2014 (e, j). The black dot denotes the areas where the trend is statistically significant (p < 0.05). The units are µg m$^{-3}$ for PM$_{2.5}$ (a,b) and ppbv for O$_3$ in (f, g), and µg m$^{-3}$ per decade (µg m$^{-3}$ decade$^{-1}$) for PM$_{2.5}$ trends (c,d,e), and ppbv per decade (ppbv decade$^{-1}$) for O$_3$ trends (h,i,j).**

From Figure 2, we also find that both PM$_{2.5}$ and O$_3$ showed a statistically significant increasing trend all over the
country from 1995 to 2014, with a nation-wide increasing rate of 6.71 µg m$^{-3}$ decade$^{-1}$ (p < 0.01) for pop-weighted PM$_{2.5}$ and 7.08 ppbv decade$^{-1}$ (p < 0.01) for pop-weighted O$_3$, respectively (Fig. S8), which was mainly driven by rapid industrialization and substantial economy development (Pandey et al., 2014; Sadavarte and Venkataraman, 2014). However, distinct spatial heterogeneity for the increasing trend was observed for the two air pollutants. The PM$_{2.5}$ exhibited varying degrees of increase across India, with the most distinctive increase occurring in the IGP, where the maximum trend reached
12.60 µg m$^{-3}$ decade$^{-1}$. This notable rise can be attributed to the increased regional ANTHRO emissions (Fig. S7). For O$_3$,





eastern central India experienced the highest $O_3$ increases, with an obvious increase in the eastern and the lowest increases in western India. One thing needs to be pointed out that in northern IGP, including New Delhi, significant $O_3$ decreases were also observed, which could be caused by the inhibited $O_3$ production due to NO titration as a result of dramatic increase in NOx emissions, as discussed in Karambelas et al. (2018). Splitting the trend into two periods (from 1995 to 2004 and from 2005 to 2014), we found a larger increasing trend in the latter period than that in the previous one for both $PM_{2.5}$ and $O_3$, which may be due to the rapid urbanization and growing transportation activities over populous regions (Fig. S9) in recent years in India (Gao et al., 2018).

### 3.3 Driving factor analysis for the air pollution changes in India

### 3.3.1 Contributions to the annual and seasonal trends

Figure 3 shows the contributions of ANTHRO and BB emissions changes on area-weighted $PM_{2.5}$ and $O_3$ trends from 1995 to 2014. Not surprisingly that ANTHRO emission changes dominate the $PM_{2.5}$ and $O_3$ deterioration in India. Changes in ANTHRO emissions alone increased area-weighted concentration of $PM_{2.5}$ by 5.46 μg m$^{-3}$ decade$^{-1}$ ($p < 0.01$) and area-weighted concentration of $O_3$ by 2.71 ppbv decade$^{-1}$ ($p < 0.01$), accounting for 102.21 % and 104.11 % of the total changes, respectively. The contributions of changes in BB emissions were relatively minor, with distinct interannual variations. Spatially, we find that both the long-term $PM_{2.5}$ and $O_3$ trends are mostly dominated by the ANTHRO emission changes all over India (Fig. S10a, c). Changes in BB emissions lead to a slight increasing trend of $PM_{2.5}$ in most of India and a decreasing trend in eastern India, though neither of these trends is statistically significant. BB emissions seem to increase $O_3$ in IGP and central India and decrease $O_3$ in western India, but the trends are insignificant either (Fig. S10b, d).



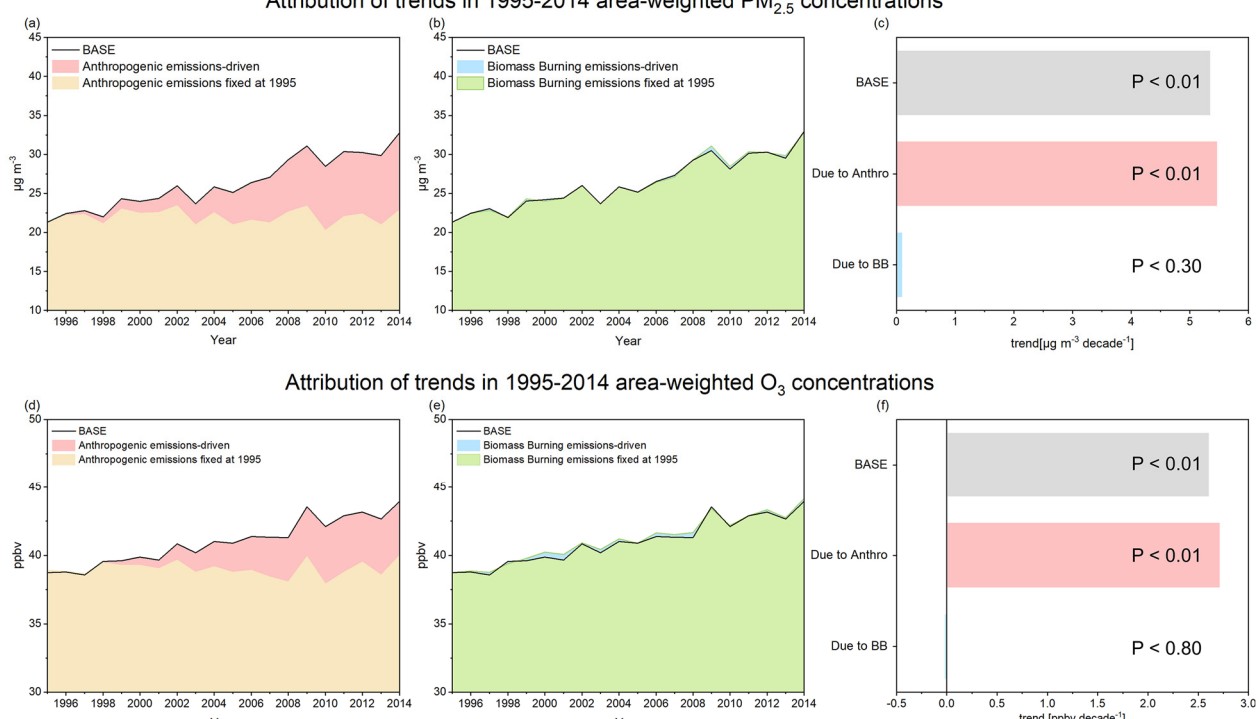

Figure 3. Drivers for trends of area-weighted (a-c) PM$_{2.5}$ and (d-f) O$_3$ in India in 1995-2014. The yellow shadings in (a, d) show the evolution of model-simulated PM$_{2.5}$ and O$_3$ concentrations in the FixAN simulation, with the red shadings illustrating the estimation of the PM$_{2.5}$ and O$_3$ concentrations resulting from changes in ANTHRO emissions compared to the 1995 level. (b, e) as for (a, d), but for impacts of changes in BB emissions. (c, f) denotes the estimated PM$_{2.5}$ and O$_3$ trends in India derived from the BASE simulation and impacts of ANTHRO and BB emissions, respectively.

It is well recognized that BB emissions usually feature a distinct seasonal trend, especially in India, where they are influenced by the monsoon. Hence, here we quantified the seasonal trend of PM$_{2.5}$ and O$_3$ from ANTHRO and BB emissions for DJF (December-January-February), MAM (March-April-May), JJA (June-July-August, monsoon season), and SON (September-October-November, post-monsoon season) from 1995 to 2014. From Fig. 4a-h, we find that the contributions of ANTHRO had consistent spatial patterns for the seasonal PM$_{2.5}$ trend, with larger influences in the post-monsoon seasons (DJF and SON), which was estimated to be responsible for PM$_{2.5}$ enhancement by as high as 17.08 μg m$^{-3}$ decade$^{-1}$ because of decreased vertical dispersion and diffusion of aerosol caused by lower solar radiation during winter and surface wind speeds (Bran and Srivastava, 2017). The contributions of ANTHRO emissions during the MAM and JJA were modulated as a result of increased precipitation, strong air convergence, and uplift strong air convergence during the presence of the summer monsoon, which impeded the accumulation of PM$_{2.5}$ concentrations at ground level (Bran and Srivastava, 2017; Gao et al., 2020; Lu et al., 2018). Unlike PM$_{2.5}$, the contributions of ANTHRO changes on surface O$_3$ trend in India had a distinct spatial pattern across seasons (Fig. 4i-p). The ANTHRO had a much stronger positive influence on the O$_3$ increases in northern, eastern central, and eastern India during JJA and SON, while it had the largest increases in southern India in the pre-monsoon season (MAM, Fig. 4j). It was reported that the stronger solar radiation and higher temperature in MAM were





attributed to an increase in the photochemical efficiency of $O_3$ in the presence of NOx (Doherty et al., 2013; Jacob and

Winner, 2009; Pusede et al., 2015). The decreased $O_3$ in the IGP was most pronounced in the DJF season (Fig. 4i), which

was mainly attributed to lower solar radiation and titration of $O_3$ by higher NOx levels (Kumar et al., 2012). Additionally,

the occurrence of winter monsoon led to extensive air subsidence in northern India, resulting in low net $O_3$ production and

strong horizontal export, which ultimately leads to relatively low $O_3$ levels (Lu et al., 2018).





Figure 4. Seasonal patterns of (a-d) ANTHRO and (e-h) BB emissions contributions for the trends of PM$_{2.5}$ in India from 1995 to 2014 and (i-p) for O$_3$. The units are µg m$^{-3}$ per decade and ppbv per decade.



### 3.3.2 Contributions to the seasonal air quality changes

Figure 5 shows the spatial distributions of BB contributions for seasonal $PM_{2.5}$ and $O_3$ changes between 1995 and 2014, respectively, as detailed in Table S3. The changes in BB emissions from 1995 to 2014 contributed significantly to the $PM_{2.5}$

increases in eastern India (over 20 µg m$^{-3}$) with a high incidence of forest fires (Jena et al., 2015). It also resulted in an increase of $O_3$ by more than 4 ppbv in eastern India in MAM. Contributions to seasonal $PM_{2.5}$ and $O_3$ changes from BB were comparable or even exceeding the those from ANTHRO in some regions, such as Manipur and Nagaland (Fig. S11). With a higher BB fraction in other years, such as 1999, these contributions could even be even higher, reaching up to 46.03 µg m$^{-3}$ and 6.46 ppbv for $PM_{2.5}$ and $O_3$, respectively (Fig. S12). Therefore, we conclude that the BB emissions in India poses a great

threat to the air quality and thus cannot be overlooked.

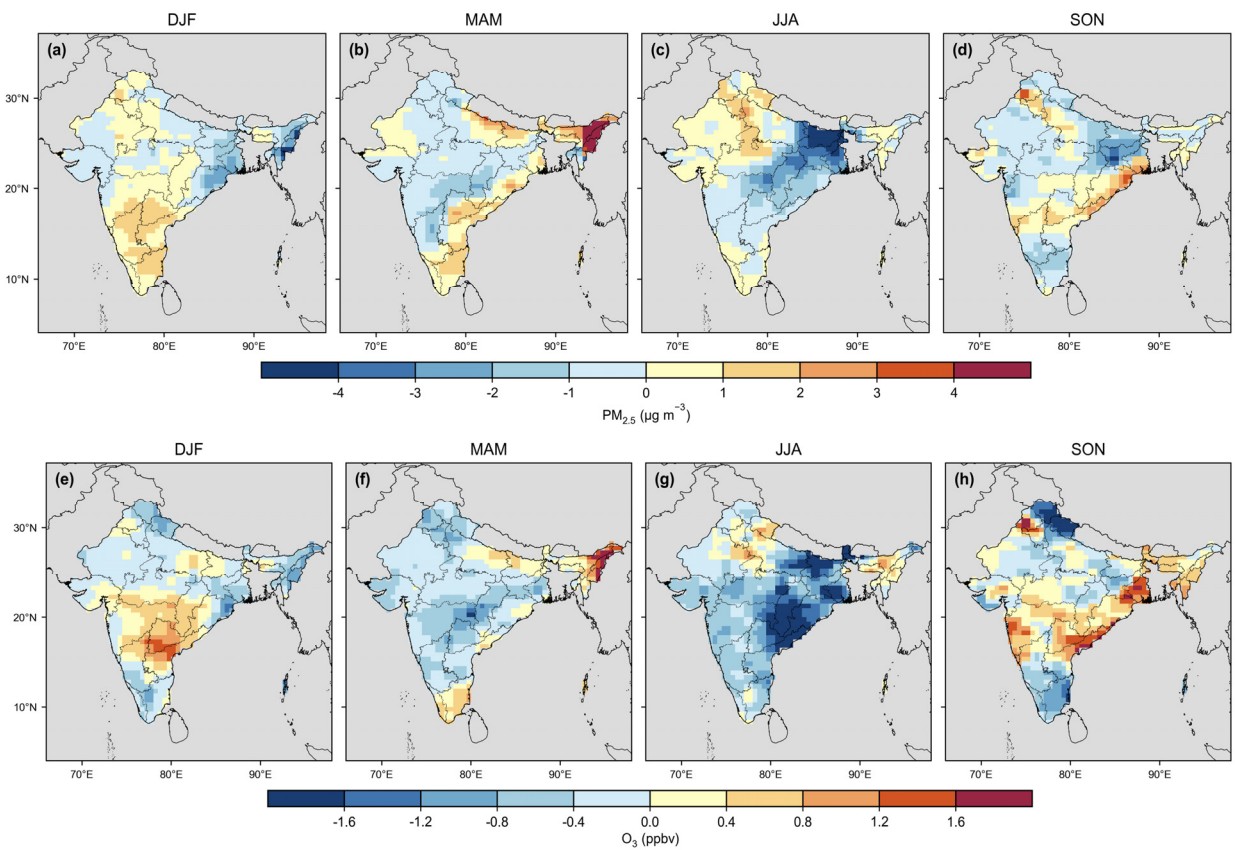

**Figure 5. Spatial distributions of the BB contribution for seasonal (a-d) PM₂.₅ and (e-h) O₃ changes from 1995 to 2014 for DJF, MAM, JJA, and SON. The contributions from BB were calculated as the differences between BASE and FixBB in 2014. The units are µg m⁻³ and ppbv.**

### 3.4 Long-term trends of premature mortality due to PM₂.₅ and O₃ in India

We estimated that the national mortality burden attributable to ambient $PM_{2.5}$ exposure rose significantly from 698.29 thousand in 1995 to 893.33 thousand in 2014, at a rate of 97.83 thousand per decade (p < 0.01, Figure 6a). Similarly, the





mortality burden attributable to O₃ exposure also notably rose from 414.50 thousand in 1995 to 580.03 thousand in 2014, being 73.91 thousand per decade (p < 0.01). We observed that the hotspots of premature mortality attributable to $PM_{2.5}$ and

O₃ exposure occurred in New Delhi and IGP regions in 1995 and 2014 (Fig. 6b-e), coincidently with the dense population (Fig. S9). We found that Uttar Pradesh, Bihar, West Bengal, and Haryana, four states within the IGP region, accounted for 41.00 % and 39.77 % of the national premature mortality due to $PM_{2.5}$ and O₃ in 2014, respectively. Considering this heterogeneous spatial distribution, it is imperative for the IGP region to implement stronger air pollution control policies to safeguard human health, as discussed in Jia et al. (2021). Our estimations for the O₃-related mortality burden were higher

than those reported from the GBD2019 (Fig. S13) since we applied a higher RR and used larger baseline mortality rates (see Methods section 2.4). After recalculating the O₃-related mortality burden using the GBD2019 metrics, we reported an increasing trend of 29.74 thousand deaths per decade$^{-1}$ for O₃-related mortality, comparable to the GBD2019 estimation of 33.24 thousand deaths per decade$^{-1}$. However, our estimated mortality burdens are still slightly higher than the GBD2019 due to the O₃ overestimation in our model (Fig. 1 and Fig. S5).

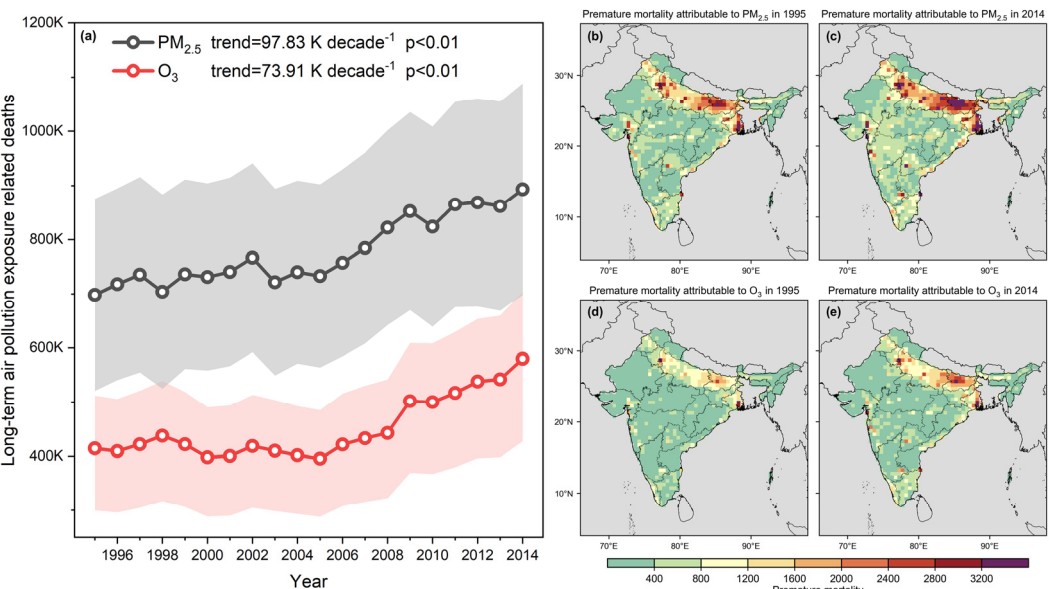


**Figure 6. Spatial-temporal change of mortality burden attributable to $PM_{2.5}$ and O₃. (a) interannual variation from 1995 to 2014. The shaded area indicates the range of 95 % confidence interval (gray indicates half of the range). (b-e) spatial distributions of the average annual premature mortality attributable to (b-c) $PM_{2.5}$ and (d-e) O₃ in 1995 and 2014.**

        To isolate the effects of population heterogeneous among regions, we also quantified the mortality burden changes per

capita (avoided deaths per 100,000 people) from 1995 to 2014 (Fig. 7). $PM_{2.5}$-attributable premature mortality per capita was higher in the IGP and eastern India, with the highest in Chandigarh (427.2), followed by Sikkim (153.6), Meghalaya (140.3), and NCT of Delhi (126.1) in 1995 (Fig. S14). The spatial distribution of O₃-attributable premature mortality per capita resembled that of $PM_{2.5}$, but the values are relatively smaller, with the maximum value also appearing in Chandigarh (288.0), followed by Sikkim (120.2), Meghalaya (68.6), and NCT of Delhi (68.0) in 1995 (Fig. S14). Over the period from 1995 to



2014, PM$_{2.5}$- and O$_3$- attributable premature mortality per capita decreased in the north and increased in the south (Fig. 7), indicating that the increasing trend of premature mortality attributable to PM$_{2.5}$ and O$_3$ in the IGP region was mainly driven by the increased population (Fig. S9).

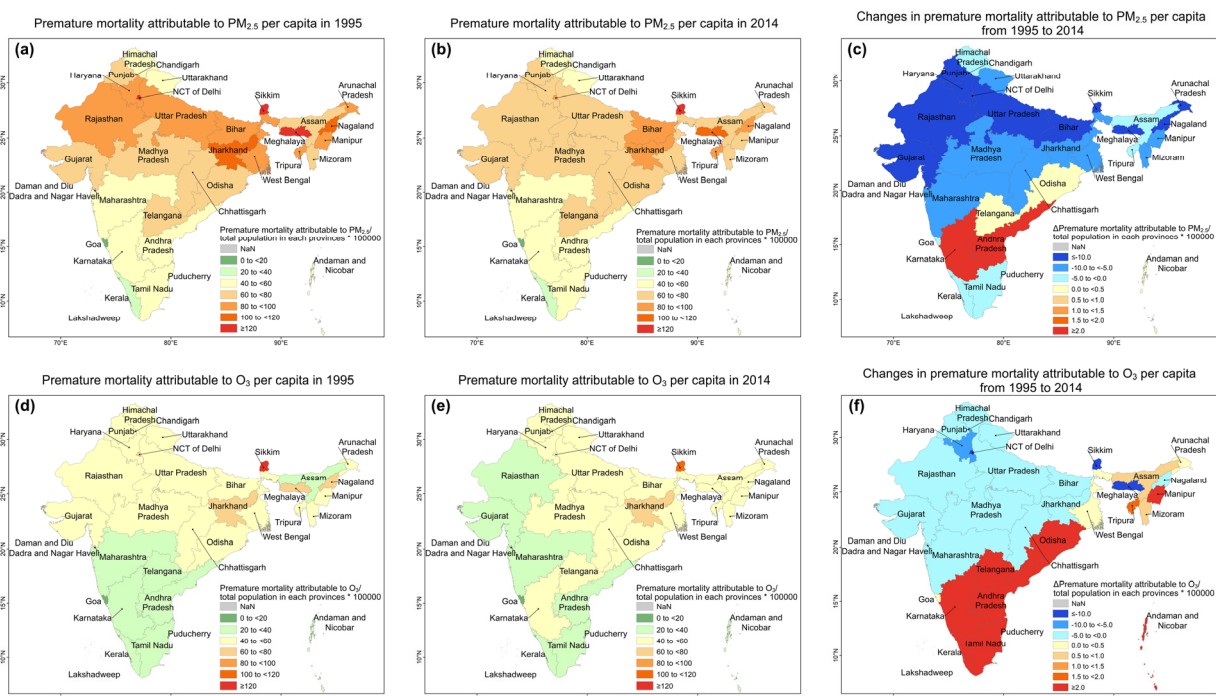

**Figure 7. Spatial distributions of premature mortality attributable to PM$_{2.5}$ or O$_3$ per capita (avoid deaths per 100,000 people) in**
**(a, d) 1995, (b, e) 2014, and (c, f) changes from 1995 to 2014 in the state of India.**

Figure 8 shows that changes in ANTHRO emissions from 1995 to 2014 increased premature mortality per capita attributable to PM$_{2.5}$, with the higher values located mainly in eastern IGP and central India. Changes in BB emissions increased premature mortality attributable to PM$_{2.5}$ per capita in eastern, western, and southern India and decreased in IGP and central India. The state with the largest increase was Manipur (2.55), followed by Nagaland (2.06), which was associated

with the high incidence of wildfires in these regions. The state that experienced the largest decrease was Jharkhand (-1.71), with Bihar (-1.02) followed behind. In order to explore contribution changes from ANTHRO and BB emissions, we estimated the premature mortality attributable to PM$_{2.5}$ per capita in 2000, 2005, and 2010-2014 in Table S4, respectively. There was a sharp rise in contributions to premature mortality attributable to PM$_{2.5}$ from changes in ANTHRO emissions from 1995 to 2014. Not surprisingly the premature mortality attributable to PM$_{2.5}$ from changes in BB emissions fluctuated

greatly from 1995 to 2014. In 2000, a year with high BB emissions (Fig. S8), the contributions of changes in BB emissions to the premature mortality attributable to PM$_{2.5}$ in the states of Mizoram, Nagaland, Arunachal Pradesh, and Tripura reached 5.14, 4.90, 4.86, and 4.17, respectively, which exceeding the contributions of changes in ANTHRO emissions in that year (Table S4).





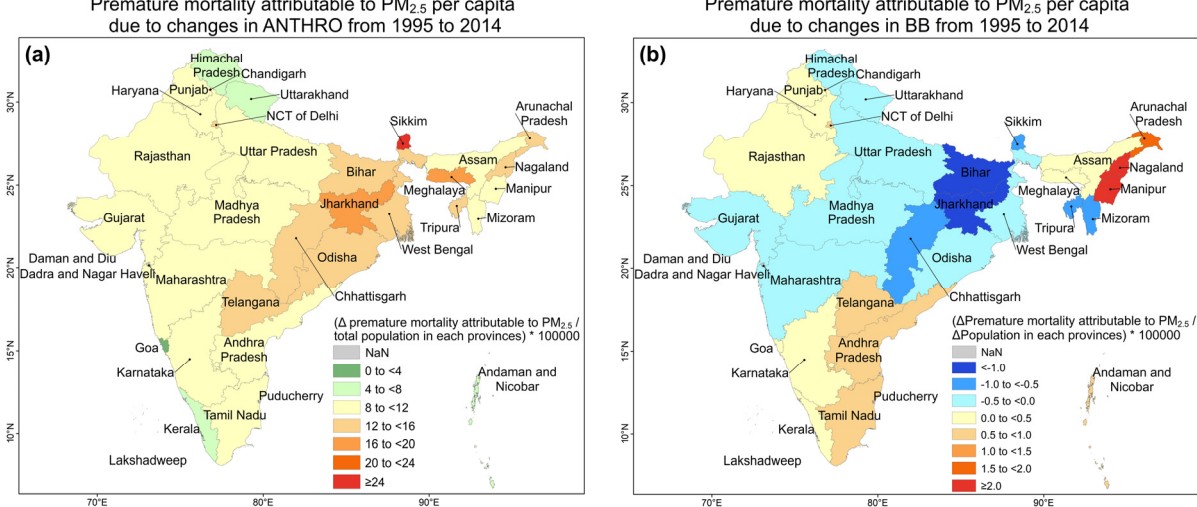

**Figure 8. Spatial distributions of contributions to premature mortality attributable to $PM_{2.5}$ per capita (avoided deaths per 100,000 people) from changes in (a) ANTHRO and (b) BB emissions from 1995 to 2014.**

## 4 Conclusions

In this study, we applied a state-of-the-art global CTM (CAM-chem) to provide a detailed assessment of long-term trends of the ambient $PM_{2.5}$ and $O_3$ in India and their health burden from 1995 to 2014, as well as the driving factor analysis from anthropogenic (ANTHRO) and biomass burning (BB) emission changes. The annual mean area-weighted $PM_{2.5}$ over India increased at 5.34 $\mu g\ m^{-3}$ decade$^{-1}$ ($p < 0.01$) from 1995 to 2014, dominated by the ANTHRO emissions (5.46 $\mu g\ m^{-3}$ decade$^{-1}$, $p < 0.01$). The highest and fastest $PM_{2.5}$ growth was in the IGP regions due to the rapid industrialization, urbanization, and transportation growth. For annual mean area-weighted $O_3$, the increase was 2.60 ppbv decade$^{-1}$ ($p < 0.01$), dominated by the ANTHRO emissions as well (2.71 ppbv decade$^{-1}$, $p < 0.01$). We found that $O_3$ concentrations were highest in northern India, with the fastest growth occurring in northern, central, and eastern India. The contributions from BB emissions for the long-term trends were not significant for either $PM_{2.5}$ (0.09 $\mu g\ m^{-3}$ decade$^{-1}$, $p < 0.30$) or $O_3$ (-0.01, $p < 0.80$), and also showed significant seasonal variations due to large inter-annual variability features. However, when we examine the air quality changes in specific years, such as 1999 and 2014, we found that the contributions from BB could be comparable to or even exceed those from ANTHRO during winter (December-January-February) and spring (March-April-May), reaching over 46.03 $\mu g\ m^{-3}$ and 6.46 ppbv for $PM_{2.5}$ and $O_3$, respectively.

Further estimation of mortality burden showed a 27.93 % (698.29 to 893.33 thousand) increase in premature mortality attributable to $PM_{2.5}$ between 1995 and 2014 (22.94 % for 2005-2014), and a 39.93 % (414.50 to 580.03 thousand) increase for $O_3$ (44.54 % increasing during 2005-2014). Changes in ANTHRO and BB emissions were responsible for an enhancement of premature mortality attributable to $PM_{2.5}$ by 88.78 % (97.83 thousand per decade, $p < 0.01$) and 0.02 % (2.38 thousand per decade, $p < 0.10$). After removing the effect of population growth, our analysis revealed a notable higher



mortality burden per capita attributable to $PM_{2.5}$ in the IGP regions. However, it is noteworthy that the mortality burden per capita in these regions exhibited a significant decline over the period of 1995-2014, despite the increasing trend of premature mortality. This suggests that population growth is the primary factor driving the trend of premature mortality.

Our study is subject to several uncertainties and limitations. First of all, the coarser resolution in the global model (0.9°
× 1.25°) is frequently found to be unable to realistically represent the complex physical and chemical processes of regional-scale air pollution, especially for $O_3$ (Yue et al., 2023). Moreover, missing chemical mechanisms in the model, such as the lack of representations of nitrate and ammonium (Ren et al., 2023) and the secondary organic aerosol (Liu et al., 2021), prevents the model from accurately simulating $PM_{2.5}$ concentration, especially during heavily polluted regions, such as China and India (Turnock et al., 2020). Another major uncertainty originates from the inaccurate emission inventory, especially for
developing regions in early periods, as reported by the global datasets (Paulot et al., 2018; Wang et al., 2022). Zhang et al. (2021b) revealed that model performance with global CEDS inventory tends to predict lower bias for surface $PM_{2.5}$ and higher bias for surface $O_3$ compared with a regional emission inventory (MEIC) in China due to disparities in spatial allocation. Xie et al. (2024) also highlighted a significant underestimation of agricultural fires in the inventory. Moreover, the uncertainty from health functions ranging from the choice of the exposure-response functions (Ostro et al., 2018; Giani et
al., 2020) and the uncertainties of the baseline mortality rates both have different impacts on human health (Lelieveld et al., 2015; Pozzer et al., 2023). Finally, other limitation included in our experimental design was that we set global fixed emissions for both anthropogenic and biomass burning instead of in India only, resulting in ignoring the impact of intercontinental transportation.

**Code and data availability**

The CESM model code is available at https://www.cesm.ucar.edu/models/cesm2/release_download.html (last access: 12 March 2024). Observation data is available at https://wustl.box.com/s/79pfex658crbq4dykxh51vvfdpksfhj5 (last access: 29 March 2024), which is collected by the Atmospheric Composition Analysis Group (ACAG) at Washington University in St. Louis.

**Author contributing**

B.L. analyzed the simulation results and wrote the manuscript. Y.Z. and T.T. conceived the idea, designed and conducted the experiment. Y.Z., T.T., H.Z., J.H., J.M., W.W., and L.X. revised the original paper. All authors contributed to the manuscript.

**Competing interests**

The authors declare that they have no conflict of interest.



## Acknowledgments

This work was supported by the National Natural Science Foundation of China (42375172). We would like to acknowledge the Atmospheric Composition Analysis Group, the Washington University in St. Louis (WUSTL), for providing open access to the satellite-derived $PM_{2.5}$ (https://sites.wustl.edu/acag/datasets/surface-pm2-5/, last accessed on March 18, 2024). We also acknowledge the High Performance Computer resources (2023-EL-PT-000184) from the National Key Scientific and Technological Infrastructure project "Earth System Numerical Simulation Facility" (EarthLab). This material is based upon

work supported by the National Center for Atmospheric Research, which is a major facility sponsored by the NSF under cooperative agreement no. 1852977. We thank all the scientists, software engineers, and administrators who contributed to the development of CESM2.

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
