# Peer review of "Spatial-temporal patterns of anthropogenic and biomass burning contributions on air pollution and mortality burden changes in India from 1995 to 2014"

_EGUsphere, 2024_

## Author Response (AR2)

**Response to Editors and Reviewers**

We gratefully thank the reviewers for the constructive comments and suggestions to improve the manuscript. As detailed below, the reviewers' comments are shown in *black italic*; our response to the comments is in blue. New or modified text is in red.

**Referee 1:**

*The authors have studied the contribution of anthropogenic and biogenic emission change between 1995-2014 to $PM_{2.5}$ and $O_3$ concentration in India. I have the following comments:*

**Response:** Thanks for the reviewer's comments and suggestions. We have addressed the specific comments and revised the manuscript accordingly. For clarity, the reviewer's comments are listed below in *black italic*, while our responses and changes in the manuscript are shown in blue and red, respectively.

1. *The maps of the Indian subcontinent don't seem to be right. If a region is disputed the authors can use dotted lines to represent it.*

**Response:** Thanks for the reviewer's comments regarding the maps of the Indian subcontinent. In our revised manuscript, we have used dotted lines to clearly indicate the disputed regions on the maps (see the plot below as example). We have made necessary changes to all the plots in our manuscript.

[Figure]

2. *In equation 1 of Theil-Sen estimator, the authors should clarify that $x_i$ and $x_j$ represent points from either $PM_{2.5}$ or $O_3$ or premature mortality. The sentence used now creates confusion that i and j might represent concentrations/premature mortality from different parameters.*

**Response:** Thanks for the reviewer's suggestion. In the revised manuscript, we have clarified that $x_i$ and $x_j$ represent the concentrations of either $PM_{2.5}$, $O_3$, or attributed premature mortality at different times $t_i$ and $t_j$, but for the same parameter. This revision ensures that the distinction between parameters is clear and avoids misunderstanding.

Line 124: Where $x_i$ and $x_j$ represent the concentrations of either $PM_{2.5}$, $O_3$, or attributed premature mortality at time $t_i$ and $t_j$ (i > j), respectively, for the same parameter. *Slope* > 0 indicates an increasing trend; *Slope* < 0 indicates a decreasing trend.

3. *The authors indicate that they use integrated exposure response function to estimate risk due*

*to PM$_{2.5}$ exposure, however they don't mention the contrafactual concentration used to estimate the risk. Does the contrafactual concentration used change over the years? Since the PM$_{2.5}$ and O$_3$ concentration change over the years in India, the contrafactual concentration used to estimate the risk should also change over the years else the risk estimated might be over or underestimated.*

**Response:** Thanks for the reviewer's comments. However, in our study, we did not consider the changes of contrafactual concentration for either PM$_{2.5}$ or O$_3$, which were consistent with previous studies, such as the one published in *Nature (*Anenberg et al., 2017*),* and how the researchers were doing in performing health impact assessments. As a matter of fact, the community has been debating whether there was a contrafactual concentration for either PM$_{2.5}$ or O$_3$, since several studies have been found significant health risks at even lower concentrations (Di et al., 2017; Sigsgaard & Hoffmann, 2024). Based on the function, without considering the contrafactual concentration, the premature mortality burden from ambient air pollution will be much bigger.

References:

*Anenberg, S. C., Miller, J., Minjares, R., Du, L., Henze, D. K., Lacey, F., Malley, C. S., Emberson, L., Franco, V., Klimont, Z., and Heyes, C.: Impacts and mitigation of excess diesel-related NOx emissions in 11 major vehicle markets, Nature, 545, 467–471, https://doi.org/10.1038/nature22086, 2017.*

*Di, Q., Wang, Y., Zanobetti, A., Wang, Y., Koutrakis, P., Choirat, C., Dominici, F., and Schwartz, J. D.: Air pollution and mortality in the medicare population, N. Engl. J. Med., 376, 2513–2522, https://doi.org/10.1056/NEJMoa1702747, 2017.*

*Sigsgaard, T. and Hoffmann, B.: Assessing the health burden from air pollution, Science, 384, 33–34, https://doi.org/10.1126/science.abo3801, 2024.*

4. *In line 200, how can ANTHRO contribute to above 100% increase in PM$_{2.5}$ and O$_3$ concentration?*

**Response:** Thanks for the reviewer's comment. The trends in PM$_{2.5}$ and O$_3$ trends are influenced by a combination of factors, including ANTHRO emissions, BB emissions, and meteorological conditions. The reason why the contributions of ANTHRO emissions to the increasing trends in PM$_{2.5}$ and O$_3$ exceeds 100% is that, the other factors, such as BB emissions and meteorological conditions, may have mitigated the growth of PM$_{2.5}$ and O$_3$. Therefore, it is reasonable that the contributions of ANTHRO emissions to the trends in PM$_{2.5}$ and O$_3$ exceeds 100%.

5. *As per figure 3, while the annual biomass burning contribution to total PM$_{2.5}$ and O$_3$ concentration remain lower I have 2 observations:*
    a) *PM$_{2.5}$ concentration due to burning should at least increase during the burning season i.e. March-May and Sep-Nov, the plot 3b doesn't capture it.*
    b) *O$_3$ concentration due to burning in plot 3e have large increases in some years whereas the increase doesn't seem much on other years. What's the reason for this yearly variability?*

**Response:** Thanks for the reviewer's constructive and insightful comments.

a) The time series in Figure 4b (corresponds to Figure 3 in the original manuscript) presents annual averages for India. The occurrence of BB is influenced by both time and location, which is why the effects of the burning season are not captured in this figure. From Figure 6a-d, we can conclude that the contribution of burning on $PM_{2.5}$ were pretty high in the two fire seasons. To make it more clearly, we changed the following sentence in the revised manuscript:

Line 227: The contributions of changes in BB emissions were relatively minor, with distinct interannual and seasonal variations.

b) The yearly variability is attributed to significant interannual variations in BB emissions, which are influenced by changing meteorological conditions and so on. This variability subsequently affects the annual contributions of BB emissions to $O_3$ concentrations.

6.  *What does the dots in figure 4 in the $O_3$ plot indicate?*

**Response:** Thanks for the reviewer's insightful comment. The dots in the $O_3$ plot of Figure 5 (corresponds to Figure 4 in the original manuscript) indicate the grid points where the trends are statistically significant with $p < 0.05$. We have added a description of the dots in the revised manuscript to clarify their significance. Please see the new legends blow. Additionally, we acknowledge an error in the representation of plots 4a-h in Figure 5, where the significance for $PM_{2.5}$ trends were left, and an error for the units. We have corrected this in the revised manuscript to ensure accurate representation of the data.

[Figure]

**Figure 5** Seasonal patterns of (a-d) ANTHRO and (e-h) BB emissions contributions to the trends of PM$_{2.5}$ in India from 1995 to 2014, and (i-p) for O$_3$. The units are μg m$^{-3}$ per decade for PM$_{2.5}$ and ppbv per decade for O$_3$. The dots in the plots indicate statistically significant trends, with p-values less than 0.05.

7. *How are the seasonal differences in PM$_{2.5}$ and O$_3$ concentration in Figure 4 &5 estimated?*

*Are they estimated as the average of the difference over the years from 1995-2014 with respect to base year 1995? Were there any years with notable increase in anthropogenic or biogenic emissions?*

**Response:** Thanks for the reviewer's insightful comment.

In Figure 5 (corresponds to Figure 4 in the original manuscript), the contributions of changes in ANTHRO and BB emissions to the seasonal trends of $PM_{2.5}$ and $O_3$ are derived by subtracting the FixAN or FixBB scenarios from the BASE scenario. The annual trends for $PM_{2.5}$ and $O_3$ across different seasons are subsequently estimated using the Theil-Sen estimator and the Mann-Kendall test, providing a robust analysis of seasonal variability.

Figure 6 (corresponds to Figure 5 in the original manuscript) evaluates the contributions of changes in BB emissions to the annual average concentrations of $PM_{2.5}$ and $O_3$ over the period from 1995 to 2014. This assessment is conducted by subtracting the FixBB in 2014 from the BASE scenario for the same year. It is crucial to emphasize that this method does not represent the average of the difference over the years from 1995-2014 with respect to base year 1995, instead, it reflects the impacts of BB emissions changes between 1995 and 2014.

As illustrated in Figure S5, there is a discernible upward trend in ANTHRO emissions year on year. Significant increases in BB emissions were observed in 1999, 2006, and 2009. We have acknowledged in the manuscript that with a higher BB fraction in other years, such as 1999, these contributions could even be even higher (Line 265).

8.  *The authors need to check for grammatical errors throughout the manuscript.*

**Response:** Thanks for the reviewer's comments. We have now checked the grammar thoroughly for the revised manuscript.

**Response to Editors and Reviewers**
We gratefully thank the reviewers for the constructive comments and suggestions to improve the manuscript. As detailed below, the reviewers' comments are shown in *black italic*; our response to the comments is in blue. New or modified text is in red.

**Referee 2:**

*The authors conducted a systematic study on the sources and health impacts of air pollutants $PM_{2.5}$ and $O_3$ in India, one of the most polluted countries. It would be very beneficial for air pollution control if done properly after addressing the major comments below covering the research scope, methods, and result interpretations.*

**Response:** Thanks for the reviewer's comments and suggestions. We have addressed the specific comments and revised the manuscript accordingly. For clarity, the reviewer's comments are listed below in *black italic*, while our responses and changes in the manuscript are shown in blue and red, respectively.

*Major comments*

*1.   Research scope.*

*The authors claim that "...but no systematic assessment has been made to investigate the individual contributions of anthropogenic and biomass burning emissions." However, after a quick search, many are published already such as, https://www.science.org/doi/epdf/10.1126/sciadv.abm4435 and https://www.sciencedirect.com/science/article/pii/S1352231016304630. The authors should do a more thorough literature review in India and report both observation-based and model-based results and refine their claim, possibly highlighting the extension to non urban areas. Particularly, explain upfront what is GBD2019 and what have they found and what is not in GBD2019 that is first studied here.*

**Response:** Thanks for the reviewer's comments. We have refined our claims in the revised manuscript and added a more thorough review of existing literature highlighting the extension to non-urban areas. Additionally, we have clarified what GBD2019 is, summarizing its findings and highlighting the aspects that are novel in our study. The modified results are as follows:

Line 14: Anthropogenic (ANTHRO) and biomass burning (BB) emissions are the major sources of ambient air pollution. India has experienced a dramatic deterioration in air quality over the past few decades, but no systematic assessment has been made to investigate the individual contributions of ANTHRO and BB emissions changes over the long term in India, extension to non-urban areas.

Line 37: The latest Global Burden Disease (GBD2019) study, a comprehensive research initiative that quantifies health loss due to diseases, injuries, and risk factors worldwide, estimated that exposure to air pollution, including both household and ambient pollution, led to 6.7 million premature deaths (95 % confidence interval [CI], 5.9 to 7.5 million) worldwide in 2019 (GBD 2019 Risk Factors Collaborators., 2020).

We add the following sentence after line 51 to clarify what is new in our study compared with

GBD2019:

Line 51: (Pandey et al., 2021). However, the GBD2019 study did not separate the air quality changes due to various contribution factors, such as anthropogenic (ANTHRO) and biomass burning (BB).

Line 67: Previous studies have utilized observational and satellite data to assess the impacts of ANTHRO and BB sources on air quality trends in some Indian cities (Gurjar et al., 2016; Vohra et al., 2022). Additionally, model simulations have been employed to analyze source contributions to air pollution(Conibear et al., 2018a, b). However, there remains a lack of comprehensive assessments regarding the impacts of long-term ANTHRO and BB emission changes on air quality, particularly in non-urban areas.

*Also, the two decades model simulation seems not fully exploited given the efforts to run the models in the first place. In the current work, only two sources are separated and their magnitude is very different. From emission data only, the results are sort of expected. What sectors are included in the current anth? Is it possible to expand the model or just analysis into specific anthropogenic sources such as industry, indoor fire use for cooking or heating, transport, etc, to provide more practical guidance for future policies and actions? Maybe use certain gas tracers from different sources or the best case scenario, they are separated in the emission data.*

**Response:** Thanks for the reviewer's comments. As the reviewer pointed out, the current simulation in our study only separates emissions into two broad categories: ANTHRO and BB emissions. This separation was chosen based on the primary goals of the study, which focused on distinguishing the major sources of emissions changes contributing to air quality impacts in India from 1995 to 2014. We were expecting the contribution of BB emissions on the surface $PM_{2.5}$ and $O_3$ changes and trends would be much larger than the results we were showing, even not comparable to the contribution from ANTHRO. As we explained in our discussion, this could be caused by the uncertainty of the BB emission we applied in our study, as well as the fact the BB is much smaller than the ANTHRO. We agree with the reviewer that further study should be carried out to distinguish the sector contributions to the surface air quality changes in India, either through tagging technique or brute force. The current simulation is already capable of spatial-temporal changes in ANTHRO and BB emissions contributions on air pollutions in India and delivering critical information for policymakers.

The anthropogenic emission inventory (CEDS version 2017-05-18) used in the study is divided into eight sectors: agriculture; energy; industrial; transportation; residential, commercial, other; solvents production and application; waste and international shipping. We add the following sentence in the revised manuscript:

Line 92: The ANTHRO emissions includes eight sectors: agriculture; energy; industrial; transportation; residential, commercial, other; solvents production and application; waste and international shipping (Hoesly et al., 2018).

In conclusion, we agree with the reviewer that more detailed separation of ANTHRO would be beneficial for studies. The current approach strikes a balance between model complexity and the provision of practical insights. The separation of ANTHRO and BB emissions already highlights the key contributors to air quality, and further refinement will be pursued in future work. For the

scope of the present study, the current results are sufficient to inform broad policy actions aimed at reducing emissions from ANTHRO and BB.

*Methods.*

1. *What are the assumptions for the Mann-Kendall test? Are they compatible with the trend method used (Theil-Sen)?*

**Response:** Thank you for the comments. The Mann-Kendall test is a non-parametric method used to identify trends in time series data, and its assumptions include:

a) **Independence of data**: The data points should be independent of each other. Our model simulation has ensured that the pollution data for each year is independent.

b) **Monotonicity**: The test detects monotonic trends, meaning it can identify whether a variable is consistently increasing or decreasing over time, which aligns with the Theil-Sen method.

Both the Mann-Kendall test and Theil-Sen estimator require independence and randomness in the data, making them suitable for identifying monotonic trends. Our simulated pollution data meets these assumptions, and the purpose is to identify monotonic trends, the two methods are compatible and provide robust support for trend analysis in air pollutants.

2. *Modeled seasonality is the only component evaluated with observations; how about the long term trend, which is more critical for this study? For the seasonality evaluation, how are the stations selected? Have the authors fully investigated the availability of all observations? Also, please show locations for all sites used.*

**Response:** Thanks for the comments. Due to a lack of long-term continuous surface observational data in India, we did not evaluate the long-term trends simulated in our study with direct observations. Instead, we utilized multi-year satellite-derived $PM_{2.5}$ data from the Atmospheric Composition Analysis Group (ACAG) at Washington University in St. Louis, as well as the bias-corrected multi-model mean of surface $O_3$ from the GBD2019 (DeLang et al., 2021) to assess long-term trends, as shown in Figures S3 and S4. We conclude that a consistent increasing trend was found in both datasets for surface $PM_{2.5}$ in India, with temporal R of 0.86 and lower estimations in our model. The model performs better in eastern India than in western India, with R usually being larger than 0.9 and NMB lower than -25 %. Similarly, compared to the simulated trend in our study with different versions of Wustl Extracts and the GBD2019, our simulated $PM_{2.5}$ concentration is lower, and the simulated $O_3$ is higher (Fig. S4).

The selection of surface observation stations was based on their availability of continuous monthly observation data. We have tried our best to collect all available surface observation results. The locations of the selected stations have been added to the revised manuscript and referred for clarity.

Line 157: We selected available ground-level $PM_{2.5}$ observations over India from previous studies (Latha and Badarinath, 2005; Panwar et al., 2013; Reddy et al., 2012; Saradhi et al., 2008; Tiwari et al., 2009, 2013), which were also collected by the ACAG. The locations of these sites are listed in Table S1.

Table S1 The locations for all observation sites.

| Site | Latitude/Longitude | Data record |
|------|--------------------|-------------|
| Anantapur | 14.62°N/77.65°E | 2010 $PM_{2.5}$ 2009 $O_3$ |
| Delhi | 28.58°N/77.20°E | 2011 $PM_{2.5}$ |
| Delhi | 28.63°N/77.18°E | 2007 $PM_{2.5}$ |
| Hyderabad | 17.47°N/78.43°E | 2003 $PM_{2.5}$ |
| Mukteshwar | 29.78°N/80.08°E | 2006-2008 $PM_{2.5}$ |
| Navi Mumbai | 19.08°N/73.00°E | 2005 $PM_{2.5}$ |
| Chennai | 13.04°N/80.23°E | 2005 $O_3$ |
| Anantapur | 8.6°N/77°E | 1997-1998 $O_3$ |
| Ahmedabad | 23.03°N/72.58°E | 2002, 2011 $O_3$ |
| Pune | 18.5°N/73.8°E | 2001-2005 $O_3$ |
| Joharapur | 19.3°N/75.2°E | 2002-2005 $O_3$ |
| Udaipur | 24.58°N/73.68°E | 2010-2012 $O_3$ |

References:

DeLang, M. N., Becker, J. S., Chang, K. L., Serre, M. L., Cooper, O. R., Schultz, M. G., Schröder, S., Lu, X., Zhang, L., Deushi, M., Josse, B., Keller, C. A., Lamarque, J. F., Lin, M., Liu, J., Marécal, V., Strode, S. A., Sudo, K., Tilmes, S., Zhang, L., Cleland, S. E., Collins, E. L., Brauer, M., and West, J. J.: Mapping Yearly Fine Resolution Global Surface Ozone through the Bayesian Maximum Entropy Data Fusion of Observations and Model Output for 1990-2017, Environ. Sci. Technol., 55, 4389-4398, https://doi.org/10.1021/acs.est.0c07742, 2021.

3. *The magnitudes of y0 and RR seem very important as the authors explain the difference compared with GBD2019 etc in Line 260 etc, changing from generally underestimated pollutants to a highly overestimated health risk. Have any sensitivity studies been done to quantify the uncertainty? Why and how are the current factors chosen? Y0 is country-dependent and age dependent only? RR is disease dependent, shouldn't it be higher in highly polluted regions such as India and IGP particularly? Explain more about these factors in addition to just showing the number.*

**Response:** Thanks for the reviewer's insightful comments. y0 here refers to the baseline mortality rates for specific cause of disease, for ozone, it referred as respiratory disease (RESP), or chronic obstructive pulmonary disease (COPD), a subset category for respiratory disease. By definition, the y0 of RESP is higher than the y0 of COPD. The GBD2017 study evaluated the ozone-related mortality burden associated with RESP, while in GBD2019 they evaluated the ozone-mortality burden associated with COPD. As we described in line 143-145 in the main context, we did quantify the uncertainty associated with it: "The recent GBD2019 reported a relatively lower *RR* for the chronic obstructive pulmonary disease (COPD), a subcategory of respiratory disease (1.06, with 95 % CI: 1.03, 1.10). To be comparable with the GBD2019 results, we also estimated the $O_3$-related mortality burden for the COPD in India during the same period."

The RR refers the relative risk, which quantifies the relation of health risk with the long-time

exposure to air pollutants. The RR will be much higher in highly polluted regions, such as China and India. For example, a recent cohort study carried out in China (Niu et al., 2022) reported a much larger RR for cardiovascular disease associated with ozone than that in US. However, based on our knowledge, no long-term cohort studies have been carried out to study the RR of specific disease with the long-term air pollution exposure. Thus, we have to use the uniformed RR value derived from the recent the GBD study. To answer the reviewer's comments, we add this as one of the limitation or suggestion in our discussion.

Line 352: Moreover, the uncertainty from health functions ranging from the choice of the exposure-response functions (Ostro et al., 2018; Giani et al., 2020) and the uncertainties of the baseline mortality rates both have different impacts on human health (Lelieveld et al., 2015; Pozzer et al., 2023). Meanwhile, in estimating the mortality burden, we applied the RR derived from a global study instead of India which could potentially have a higher value. Thus, our estimations for the air pollution-related mortality burden could be conservative. More epidemiology studies should be carried out in India to retrieve their own RR.

References:

Niu, Y., Zhou, Y., Chen, R., Yin, P., Meng, X., Wang, W., Liu, C., Ji, J. S., Qiu, Y., Kan, H., and Zhou, M.: Long-term exposure to ozone and cardiovascular mortality in China: a nationwide cohort study, The Lancet Planetary Health, 6, e496-e503, https://doi.org/10.1016/S2542-5196(22)00093-6, 2022.

2.    *Result Interpretations.*

1.    *Basic presentation. Please specify what statistical metrics are used for PM$_{2.5}$ and ozone in all figures and for all numbers. Annual mean or other? Not just PM$_{2.5}$ and Ozone. Also, the units are not even consistent. E.g. trend for Ozone and concentration for PM in Figure 4. Please check throughout the manuscript. Lastly, the colorbar/scale for Figure 2 etc could use some improvement as no spatial info can be seen for regions with highest numbers.*

**Response:** Thanks for the reviewer's insightful comments. We have specified the statistical metrics used in all figures, clarifying whether they represent annual means or other metrics. We apologize for the misleading description in Figure 5 (corresponds to Figure 4 in the original manuscript). We have ensured consistency in units throughout the revised manuscript. Additionally, we have improved the colorbar and scale for Figure 3 (corresponds to Figure 2 in the original manuscript) to enhance the visibility of spatial information for regions with the highest values. The modified plots are as follows:

[Figure]

**Figure 5.** Seasonal patterns of (a-d) ANTHRO emissions and (e-h) BB emissions contributions to the trends of PM$_{2.5}$ in India from 1995 to 2014, and (i-p) for O$_3$. The units are µg m$^{-3}$ per decade for PM$_{2.5}$ and ppbv per decade for O$_3$. The dots in the plots indicate statistically significant trends, with p-values less than 0.05.

[Figure]

**Figure 3.** Spatial distributions of $PM_{2.5}$ (top panel) and $O_3$ (bottom panel) for annual average in 1995 (a, f) and 2014 (b, g), with the trends from 1995 to 2004 (c, h), 2005 to 2014 (d, i), and 1995 to 2014 (e, j). The black dot denotes the areas where the trend is statistically significant ($p < 0.05$). The units are $\mu g\ m^{-3}$ for $PM_{2.5}$ (a,b) and ppbv for $O_3$ in (f, g), and $\mu g\ m^{-3}$ per decade ($\mu g\ m^{-3}$ decade-1) for $PM_{2.5}$ trends (c,d,e), and ppbv per decade (ppbv decade$^{-1}$) for $O_3$ trends (h,i,j).

2. *Interpretations in Section 3.3 need a lot of improvements to be readable. Explain in detail: what you are what you want to tackle; what you are showing in the figures including how you calculated the results; what did you see and what are you comparing certain things with or based on. It seems the authors are comparing different seasons rather than explaining trends (main focus) in Page 9. For Section 3.3.2, What's the point given Section 3.3.1 and Figure 4? why focus on change from 1995 to 2014 only? BB "contributed significantly"? It seems the analysis is not fair and conclusion driven given the large variation in the nature of BB emission.*

**Response:** Thanks for the reviewer's insightful comments. In Section 3.3, we aim to disentangle the contributions of ANTHRO and BB emissions to long-term trends in $PM_{2.5}$ and $O_3$ concentrations in India from 1995 to 2014. Firstly, we analyze the annual and seasonal trends for the two air pollutants, and the contributions from both ANTHRO and BB emissions. Furthermore, considering that BB emissions exhibit significant interannual variabilities, we then discussed the contributions of BB emissions to seasonal air quality changes.

Figure 4 illustrates the contributions of annual trends in area-weighted $PM_{2.5}$ and $O_3$ from ANTHRO and BB emissions. The spatial distribution of trends changes is shown in Figure S9. Figure 5 highlights the seasonal patterns of ANTHRO and BB emissions and their individual contributions to $PM_{2.5}$ and $O_3$ trends. The contribution of each emission type was assessed using sensitivity simulations, where ANTHRO or BB emissions were varied independently, holding other factors constant. Trends were quantified by comparing the full-period model simulations with the results from these sensitivity runs. The trend estimation method is described in Section 2.3.

The focus in section 3.3.1 is to investigate how changes in ANTHRO and BB emissions impacted air pollution in India. Our analysis shows that ANTHRO emissions play a dominant role in driving long-term trends, whereas the contribution of BB emissions, while smaller in magnitude, becomes significant during certain high BB years, warranting a deeper seasonal analysis.

Section 3.3.2 builds upon Section 3.3.1 and Figures 4 and 5, offering further detail by specifically addressing BB's contribution to seasonal air quality degradation. BB emissions exhibit a high degree of interannual variability, leading to less clear trends in the annual data. Thus Figure 6 focuses on the spatial distribution of BB contributions to seasonal variations in $PM_{2.5}$ and $O_3$, rather than trends. As demonstrated in Figure 6, during years with higher BB emissions, the contribution of BB to air pollution is significant, particularly in the eastern and northeastern regions of India. This warrants the focused analysis in Section 3.3.2, which explores the contributions of BB emissions during intense burning years, adding an important dimension to the overall understanding of the emission trends.

We chose to focus on the period from 1995 to 2014 as this timeframe is sufficiently long to capture meaningful changes in both air quality and emission sources, including significant variability in BB emissions. We agree that the year-to-year fluctuations in BB emissions may challenge clear trend identification. However, our analysis demonstrates that BB emissions contribute significantly to seasonal air quality deterioration. Furthermore, we have included an expanded discussion on the influence of BB in high-burning years such as 1999, providing additional context for these fluctuations (line 265).

We acknowledge the reviewer's concern of the variability in BB emissions. While BB emissions do show large year-to-year fluctuations, they contribute significantly to seasonal air quality deterioration in certain regions, particularly during peak burning years. Therefore, we conclude that BB emissions, despite their variability, pose a substantial threat to air quality, especially in eastern and northeastern India.

To avoid the confusion about the section 3.3.1 and section 3.3.2 based on two reviewers' comments, we further explain "trends" and "changes." In Section 3.3.1, we utilize "trends" to describe the contributions of ANTHRO and BB emissions to annual and seasonal air pollution trends in India, expressed in units of $\mu g\ m^{-3}$ per decade and ppbv per decade, reflecting a continuous concept. All instances of "trend/trends" in the manuscript have been derived using the method outlined in Section 2.3. Conversely, in Section 3.3.2, we use "changes" to specifically describe the contributions of BB emissions to seasonal air quality changes due to the large year-to-year fluctuations in BB emissions. "changes" refers to the differences observed from Year A to Year B, without reference to emissions in intermediate years.

3. *The conclusions have not been reviewed yet given above mentioned problems.*

**Response:** We are looking forward to the reviewer's comments on our discussions. Please see our revised version based on the two reviewers' comments.

*Minor comments*

1. *BB is used in the abstract before its full form is introduced. Check all other abbreviations as well throughout the manuscript.*

**Response:** Thanks for the reviewer's suggestion. We now defined the BB in the abstract, and we will thoroughly review the manuscript to ensure that all abbreviations, including "BB" for biomass burning, are introduced in their full form prior to their first use.

Line 14: Anthropogenic (ANTHRO) and biomass burning (BB) emissions are the major sources of ambient air pollution.

2. *Some citations are problematic. Check and correct the whole manuscript for all vague and mismatched citations. Some examples below.*

   a) *Line 31: What is WHO database? A link?*

   b) *Line 35: Is Murray et al (2020) the proper citation for GBD2019? I suspect not. Use proper citation for GBD2019 as in the ref list. Also Murray is not in the list if it is really a proper citation here.*

   c) *Line 40: IQAir is not in the reference list and with no link.*

   d) *Line 118: is Stanaway ··· the proper citation for GBD2017?*

**Response:** Thanks for the reviewer's comments. We now add the complete list for the above references:

a) We have clarified what is meant by the "WHO database" and include a link or a more detailed description as appropriate. The modified results are as follows:

Line 29: According to the World Health Organization (WHO) database, 99 % of the global population lives in areas where air quality surpasses WHO guideline limits (WHO: Air Pollution, World Health Organization, available at: https://www.who.int/health-topics/air-pollution#tab=tab_1, last access: 21 September 2024).

b) We have verified the citation for GBD2019 and ensure that the proper reference is included in the reference list. The modified results are as follows:

Line 37: The latest Global Burden Disease (GBD2019) study, a comprehensive research initiative that quantifies health loss due to diseases, injuries, and risk factors worldwide, estimated that exposure to air pollution, including both household and ambient pollution, led to 6.7 million premature deaths (95 % confidence interval [CI], 5.9 to 7.5 million) worldwide in 2019 (GBD 2019 Risk Factors Collaborators., 2020).

References:

*GBD 2019 Risk Factors Collaborators: Global burden of 87 risk factors in 204 countries and territories, 1990-2019: a systematic analysis for the Global Burden of Disease Study 2019, The Lancet, 396, 1223-1249, https://doi.org/10.1016/S0140-6736(20)30752-2, 2020.*

c) We have added IQAir to the reference list and provide a link. The modified results are as follows:

Line 43: For example, India was ranked as the most polluted country in the world in 2021, with 63 of the world's 100 most polluted cities (IQAir: 2021 World Air Quality Report, available at: https://lib.icimod.org/record/35767/files/HimalDoc2022_2021WorldAirQualityReport.pdf?type=primary, last access: 21 September 2024).

d) We have verified the citation for GBD2017 and ensure that the proper reference is included in the reference list. The modified results are as follows:

Line 138: Following our previous work (Zhang et al., 2021), we obtained the baseline mortality rate ($y_0$) for each country and 5-year age group from 1995-2014 from the GBD2017 project (GBD 2017 Risk Factor Collaborators., 2018).

References:

*GBD 2017 Risk Factor Collaborators: Global, regional, and national comparative risk assessment of 84 behavioural, environmental and occupational, and metabolic risks or clusters of risks for 195 countries and territories, 1990-2017: a systematic analysis for the Global Burden of Disease Study 2017, The Lancet, 392, 1923-1994, https://doi.org/10.1016/S0140-6736(18)32225-6, 2018.*

We appreciate your thorough review and ensure that all citations are clear, accurate, and consistent throughout the revised manuscript.

3. *"Meanwhile, the faster chemical reaction rates in India due to the strong convection, sunlight, and warm temperatures, making it a hot spot for accumulating major air pollutants compared with other regions and easily affecting the air quality in downwind regions (Zhang et al., 2016, 2021a)." doesn't make sense and please rewrite, maybe breakdown for clarity. E.g., warm temperatures seem too conservative for high ozone; strong convection contradicts accumulating.*

**Response:** Thanks for the reviewer's insightful comment. We have made the following revisions to the manuscript:

Line 52: Meanwhile, the elevated chemical reaction rates in India, driven by intense sunlight and warm temperatures, create conditions conducive to ozone formation. Additionally, strong convection enhances the transport of ozone and its precursors, such as NOy, to higher altitudes where the ozone lifetime is prolonged, facilitating accumulation. This phenomenon positions India as a hotspot for ozone pollution, significantly impacting air quality in downwind regions (Zhang et al., 2016, 2021a).

4. *Figure S1. Please include the model domain (if not global) and grid cells used for each state in Figure S1 and move it to Figure 1 as the majority of readers would want to see that. Incorporate IGP in the figure caption/figure as it is referred to a lot. Also, as topography plays an important role as well for the transport of air pollutants etc, please include topography and surrounding regions as well. Also, Section 2.3 seems too late to introduce your study region.*

**Response:** Thanks for the reviewer's suggestions. We have incorporated the model domain and the grid cells used for each state into Figure S1 and move this figure to the main text as Figure 1. Additionally, we have included references to the Indo-Gangetic Plain (IGP) in both the figure and the figure caption. To enhance the clarity of the figure, we have also included topography and surrounding regions. Furthermore, we have revised the manuscript to introduce the study region earlier by moving the sentences from line 129:

"We will discuss the air quality and mortality burden changes in six Indian regions based on meteorological conditions and aerosol variability (Fig. S1)."

After line 107:

"We will discuss the air quality and mortality burden changes in six Indian regions based on meteorological conditions and aerosol variability (Fig. 1)." The modified results are as follows:

[Figure]

**Figure 1.** A map of India marked into six regions based on meteorological conditions and aerosol variability (adapted from David et al. 2018).

5. *How was the grid size chosen as most of the listed references/model input have 0.5 degree resolution? Why is there a large difference between lat and lon resolution?*

**Response:** Thanks for questions. In this study, we utilized the CAM-chem model, a global atmospheric chemistry transport model (CTM), with a current horizontal resolution of 1.25° (longitude) × 0.9° (latitude) which is the finest at global scale. Previous studies have been utilizing regional CTM which can have fine horizontal resolution, such as 0.5 degree, 36km, or even 12km.

However, the application of the regional high-resolution CTM requires additional work to apply regional or local parameters, such as the land use or topography data. Nonetheless, subsequent model evaluation results have shown that CAM-chem effectively reproduces the spatial and temporal distribution of $PM_{2.5}$ and $O_3$ in India, which can provide valuable insights for research on air pollution in the region. We will move on to develop and apply regional CTM to study the air quality changes in regional scale for our future work.

In CAM-chem, the atmospheric and oceanic grids are not divided into square (1:1 ratio) cells due to the spherical shape of the Earth. This design helps to mitigate severe distortion and deformation issues, particularly near the poles. Here are several reasons why the latitude and longitude grid ratios differ:

a) **Earth's Shape**: The Earth is a sphere, and latitude lines (north-south) are evenly spaced circles, while longitude lines (east-west) converge at the poles. To maintain approximately equal areas for each grid cell, the latitude and longitude ratios are typically not 1:1. The longitude interval decreases as one moves toward the poles, necessitating a larger longitude interval (such as 1.25°) and a smaller latitude interval (such as 0.9°) for more uniform grid distribution.

b) **Reduction of Grid Distortion**: Using square grids would result in extreme stretching near the poles, where grid cells would be disproportionately larger than those near the equator. Non-uniform latitude and longitude grids help achieve a more even area distribution across the globe, minimizing errors in simulations.

c) **Optimization of Computational Resources**: Meteorological and climatic changes exhibit different spatial characteristics. Typically, the latitude direction shows finer features, while the longitude direction changes more smoothly. Therefore, an asymmetric grid resolution (such as 0.9° × 1.25°) optimizes simulation efficiency and accuracy, allowing for a reduced total grid count and lower computational costs.

In summary, the non-1:1 latitude and longitude grid design in CAM-chem is intended to better accommodate the spherical shape of the Earth, reduce grid distortion, optimize computational resources, and accurately capture atmospheric phenomena.

6. *A more detailed description of the emission data used. What temporal and spatial resolutions? Is there any interpolation involved? Are they evaluated? How well do they perform? Were other emission sources used in addition to Anth and BB? Line 157, mention this earlier at your model description.*

**Response:** Thanks for the reviewer's constructive comments. We have provided a more detailed description of the emission inventory in the revised manuscript. The emissions inventories employed has a spatial resolution of 0.5° × 0.5° and a temporal resolution of monthly averages. The emission inventory was interpolated from its original resolution (0.5° × 0.5°) to the target model resolution (0.9° × 1.25°) before being input into the model. We have added this clarification to the revised manuscript. The emission inventories utilized in our study have been widely adopted and validated in numerous published studies (e.g., Emmons et al., 2020; Hoesly et al., 2018). Other emission sources have been added in the revised manuscript.

Line 90: Global historical ANTHRO emissions were adopted from CEDS (version 2017-05-18), which provides monthly emissions of anthropogenic aerosol and precursor compounds at 0.5° × 0.5° from 1750 to 2014 and were used in the Coupled Model Intercomparison Project Phase 6 (CMIP6) experiments (Emmons et al., 2020; Hoesly et al., 2018). The ANTHRO emissions includes eight sectors: agriculture; energy; industrial; transportation; residential, commercial, other; solvents production and application; waste and international shipping (Hoesly et al., 2018). The air pollutants from the CEDS inventory, especially the NMVOC, were then re-speciated to match the chemical species in the latest CESM2 model, following the steps introduced by Emmons et al. (2020). Interpolation of the emission inventory from its original resolution (0.5° × 0.5°) to the target model resolution (0.9° × 1.25°) before being input into the model. Global historical BB emissions were sourced from van Marle et al. (2017) at monthly temporal resolution and 0.5° native resolution, with all emissions occurring at the surface. Additionally, the biogenic emissions were calculated using the Model of Emissions of Gases and Aerosols from Nature (MEGAN v2.1). More emissions used are described in Emmons et al., (2020).

7. *Line 84. Is year-varying even a word?*

**Response:** Thanks for the reviewer's comment. We have now revised the phrase "year-varying" to "year-to-year variability" for improved clarity. The modified results are as follows:

Line 102: The standard simulation (BASE) was driven by year-to-year variability ANTHRO and BB emissions from 1995 to 2014, as described above.

8. *Are all "trend/trends" and "rates" in the manuscript derived using the same method in Section 2.3? Are the "change/changes" only referring to change from Year a to Year b? Sometimes they are a bit confusing with the "change/changes". Please clarify. Also how did you decide where to use trends or changes.*

**Response:** Thanks for the reviewer's valuable comments. All instances of "trend/trends" in the manuscript have been derived using the method described in Section 2.3. The term "rates" appears in the context of baseline mortality rates and mortality rates per capita, which refer to the proportion of deaths within the total population. All instances of "change/changes" specifically refer to the differences observed from Year A to Year B. For example, in Figure 5, we illustrate the contribution of emission changes to the trends of $PM_{2.5}$ or $O_3$ from 1995 to 2014, expressed in units of $\mu g\ m^{-3}$ per decade and ppbv per decade, reflecting a continuous concept. In contrast, Figure 6 focuses on the changes in BB emissions contributing to $PM_{2.5}$ or $O_3$ concentrations between 1995 and 2014, without reference to intermediate years' emissions. The rationale for where to use trends or changes is that ANTHRO exhibit relatively small interannual variability and a consistent upward trend, warranting the use of "trend/trends" to describe their long-term contributions to air pollution. Conversely, BB demonstrate significant interannual variability, making "trend/trends" less statistically meaningful; thus, we opted to analyze changes from Year A to Year B.

9. *Why is there only one data point each year for the 6 mon MDA8 in figure 1b?*

**Response:** Thanks for the reviewer's comment. The data point displayed in Figure 2b (corresponds to Figure 1 in the original manuscript) represent 6 months average of MDA8 $O_3$ (6mMDA8 $O_3$) for each year from 1995 to 2014. By doing this, we can compare our results with

the GBD2019 results and previous studies who have only reported the ozone concentration in India for single year. The modified results are as follows:

Line 170: The comparisons indicate our simulated results using the CAM-chem agree very well with previous studies for both PM$_{2.5}$ and O$_3$, based on either the various metrics (such as annual average and annual maximum of 6mMDA8 O$_3$) or the population-weighted averages, consistent with the findings within the multiple CMIP6 models (Turnock et al., 2020).

[Figure]

**Figure 2.** Comparison of annual PM$_{2.5}$ and O$_3$ concentrations in India with previous studies. Note that the metrics vary depending on the study.

10.  *Line 154: Again, please show your model domain. Is the performance difference because of boundary conditions?*

**Response:** Thanks for the suggestions. We have now moved Figure S1 into the main context as Figure 1 to show our study domain. Also, in this study we used a global chemical transport model. As such, the performance of our simulations is not affected by boundary conditions.

11.  *Use proper color scale for Figure S4c.*

**Response:** Thanks for the reviewer's suggestion. We have updated Figure S3c (corresponds to Figure S4 in the original manuscript) with a more appropriate color scale. The modified results are as follows:

[Figure]

**Figure S3.** (a) Comparison of the interannual variations of surface PM$_{2.5}$ between Wustl_Extracts (reanalysis ground-level annual PM$_{2.5}$ concentrations from the Atmospheric Composition Analysis Group (ACAG) at Washington University in St.Louis.) (blue triangle) and the BASE simulation

(red circle), and spatial distribution of (b) correlation coefficient, and (c) annual mean NMB from 1998 to 2014.

*12. Figure S6c is misleading. Should not use stacked bars.*

**Response:** Thank you for your comment regarding Figure S5 (corresponds to Figure S6 in the original manuscript). We understand that the use of stacked bars may have caused confusion. After reviewing the content, we realized that Figure S5c was not directly relevant to the discussions in the manuscript. Therefore, we have decided to remove Figure S5c to avoid any potential misunderstandings. The modified results are as follows:

[Figure]

**Figure S5.** Annual emissions for (a) ANTHRO and (b) BB and in India from 1995-2014 from the CEDS.

*13. Line 170, 285: Please show a fire and anthro spatial map somewhere.*

**Response:** Thanks for the reviewer's insightful suggestion. We have included spatial maps in Figure S6 to illustrate the distributions of ANTHRO and BB emissions. Specifically, panels (a) and (b) shows that there were hotspots of ANTHRO emissions in the IGP region. Panels (d) and (e) highlight that BB emissions are predominantly found in Punjab and Eastern India. Additionally, panels (g) and (h) show that areas with a higher proportion of BB emissions are in Northern and Eastern India. We believe these visualizations will enhance the clarity of our findings.

*14. Why not use the emission trend in S7? Also, S7c does not show negative as suggested Line 170, if it is comparing the change with other regions in the first place. Consider changing the colorbar if so.*

**Response:** Thanks for the reviewer's comments. We chose not use emission trends in Figure S6 (corresponds to Figure S7 in the original manuscript) due to the significant interannual variability of BB emissions. Instead, we focused on the changes between 1995 and 2014 to illustrate both the growth of ANTHRO emissions over the past two decades and the spatial distribution variations during years of high and low BB emissions. Regarding Line 170, it does not imply that BB emissions in these areas are increasing while ANTHRO emissions are decreasing. We have made the following modifications to the manuscript:

Line 190 Unlike other administrative regions, northern and eastern India, such as Punjab and Manipur, features a higher ratio of BB emissions to ANTHRO emissions.

*15. Line 175 "Unlike". They seem similar to me.*

**Response:** Thanks for the reviewer's comments. We agree that there are certain similarities between $PM_{2.5}$ and $O_3$ distributions, as both show an increasing trend from south to north. However, upon closer examination, it becomes evident that $O_3$ concentrations also exhibit a rise from east to west. Notably, the highest $O_3$ levels are found in northern India and the eastern part of central India. To make it clear, we revised the following sentence:

Line 196: Surface annual average $O_3$ concentrations gradually increase from west to east and south to north, with the highest levels concentrated in northern India and the eastern part of central India.

*16. How large is a grid in Figure S9?*

**Response:** Thanks for the reviewer's insightful comment. The horizontal resolution of the population data in Figure S8 (corresponds to Figure S9 in the original manuscript) is 0.1°. The modified results are as follows:

Line 145: Population distribution with age stratification data ($pop$) was retrieved from the GBD2017 with a horizontal resolution of 0.1°.

*17. Line 200. What is area-weight? Spatial average?*

**Response:** Thanks for the reviewer's questions. In Line 222, "area-weighted" does refers to the spatial average of $PM_{2.5}$ or $O_3$ concentrations, by considering the varying areas of different grid cells. The formula used for the calculation is as follows:

$$Area - weighted\ average = \frac{\sum_{i=1}^{n} C_i \times Area_i}{\sum_{i=1}^{n} Area_i}$$

where $C_i$ represents the $PM_{2.5}$ or $O_3$ concentrations in region $i$, $Area_i$ denotes the area of region $i$, and $n$ is the total number of regions.

*18. Figure 6. What is the 95% CI? Spatial variability by grid?*

**Response:** Thanks for the reviewer's questions. The 95% CI shown in Figure 7 (corresponds to Figure 6 in the original manuscript) is derived from the relative risk (RR) estimates of long-term exposure to $PM_{2.5}$ and $O_3$, as described in Section 2.4.

*19. Rewrite Line 286-287 and add base year of 1995. Also, please explain why these years are*

*chosen.*

**Response:** Thanks for the reviewer's suggestion. The base year of 1995 was not added because our simulations were designed with 1995 as the BASE year, during which ANTHRO and BB emissions in 1995 did not change compared with the BASE scenario. The years 2000, 2005, and 2010-2014 were selected because the population data retrieved from the GBD2017 became continuous after 2010, while data prior to that were collected every five years. Similar results were also presented in GBD2017. To make it clear, we revised the sentence in the revised manuscript.

Line 309: To explore contribution changes from ANTHRO and BB emissions, we estimated the premature mortality attributable to $PM_{2.5}$ per capita in 2000, 2005, and 2010-2014 in Table S5, respectively, consistent with the demonstrations from GBD2017.

*20. What is the satellite-derived PM in acknowledgement??? Where was it used?*

**Response:** Thanks for the reviewer's question. The satellite-derived $PM_{2.5}$ mentioned in the acknowledgments was from Atmospheric Composition Analysis Group, the Washington University in St. Louis (WUSTL) and were used to evaluate our model results in simulating the long-term $PM_{2.5}$ trend, as illustrated in Figure S3.

---

## Author Response (AR3)

**Response to Editors and Reviewers**

We gratefully thank the reviewers for the constructive comments and suggestions to improve the manuscript. As detailed below, the reviewers' comments are shown in *black italic*; our response to the comments is in blue. New or modified text is in red.

**Responses to Referee 1:**

*Updated Figure 1 still has solid lines for all regions. Please check all the figures again.*

**Response:** Thanks for your feedback. We have corrected the solid line issue in Figure 1 and have thoroughly checked and revised all figures in the manuscript.

[Figure]

Figure 1. A map of India marked into six regions based on meteorological conditions and aerosol variability (adapted from David et al. 2018).

*Please incorporate the methodology explained in Response 7 in the revised manuscript if not already.*

**Response:** Thanks for your feedback. We have incorporated the methodology explained in Response 7 into the revised manuscript as follows:

Line 239: Hence, here we quantify the seasonal trend of $PM_{2.5}$ and $O_3$ from ANTHRO and BB emissions for DJF (December-January-February), MAM (March-April-May), JJA (June-July-August, monsoon season), and SON (September-October-November, post-monsoon season) from 1995 to 2014 by subtracting the BASE scenario from the FixAN or FixBB scenarios. The annual trends for $PM_{2.5}$ and $O_3$ for each season were subsequently estimated using the

Theil-Sen estimator and the Mann-Kendall test.

Line 266: These contributions are quantified by subtracting the BASE scenario in 2014 from the FixBB scenario in 2014.

**Responses to Referee 2:**

*Response 1. The Line 14 is not grammatically correct and did not highlight the non-urban areas properly. Please revise. Also, what did they find in those earlier studies mentioned in Line 67? Does this study validate their findings? Please include in your result discussion.*

**Response:** Thanks for your feedback. We have made the following revision:

Line 14: Anthropogenic (ANTHRO) emissions and biomass burning (BB) are major contributors to ambient air pollution, with the latter playing a particularly dominant role in non-urban regions.

Gurjar et al. (2016) demonstrated that anthropogenic sources are the primary contributors to SPM and $PM_{10}$ in three megacities in India based on observational data. And, Vohra et al. (2022) found that increases in $NO_2$ in Indian urban areas were driven almost exclusively by anthropogenic sources, not traditional biomass burning, using both observational and satellite data. Our study, which covers both urban and non-urban areas in India, supports these findings as shown by the overall trend contributions. It confirms that changes in ANTHRO emissions were the dominant factor behind the deterioration of $PM_{2.5}$ and $O_3$ in the country, as stated in Line 224. To make it clearer, we revise line 224:

Line 222: Not surprisingly, changes in ANTHRO emissions dominated the deterioration of $PM_{2.5}$ and $O_3$ in India, consistent with previous studies based on both observational and satellite data (Gurjar et al., 2016; Vohra et al., 2022).

*Response to methods. Please incorporate it in the revised manuscript if not already.*

**Response:** Thanks for your feedback. We have made the following revisions to the manuscript:

Line 125: Both the Mann-Kendall test and Theil-Sen estimator require independence and randomness in the data, making them suitable for identifying monotonic trends.

*Response to Interpretation Comment 2. Please incorporate it in the revised manuscript to improve the flow of your manuscript. Rather than start your discussions with "Figure x shows…", explain what/why you are going to do first. While you "concluded" something, the figures "show", not "showed", something, etc.*

**Response:** Thanks for your feedback. We have incorporated it into the revised manuscript and clarify the context. as follows:

Line 221: To disentangle the contributions of ANTHRO and BB emissions to long-term trends in $PM_{2.5}$ and $O_3$ concentrations in India from 1995 to 2014, we first analyze their contributions to annual and seasonal trends (Figure 4).

Line 264: BB emissions exhibit a high degree of interannual variability, leading to less clear trends in the annual data. Thus, Figure 6 focuses on the spatial distributions of BB contributions for seasonal $PM_{2.5}$ and $O_3$ changes between 1995 and 2014 rather than trends, as detailed in Table S4.

Line 272: Therefore, despite their variability, the BB emissions in India posed a great threat to the air quality and thus could not be overlooked.

*What do you mean by "to higher altitudes where the ozone lifetime is prolonged, facilitating accumulation"?*

**Response:** Thanks for your feedback. Due to strong convection, ozone and its precursors are transported to higher altitudes, where ozone tends to remain in the atmosphere for a longer period because the rate of destruction is slower. As a result, ozone can accumulate at these altitudes. Here's the revised version:

Line 53: Additionally, strong convection enhances the transport of ozone and its precursors, such as NOy, to higher altitudes, where the prolonged ozone lifetime promotes accumulation.

*For the grid choice, I understand the different lat lon for global study. However, this study conducted new simulations for the purpose of a regional study of India. Why do the authors still keep the same resolution and model domain as global modeling, especially given the availability of 0.5 degree resolution emission and other higher resolution data?*

**Response:** Thanks for your feedback. This study is based on the results of global model simulations to provide a detailed analysis of India, which is why the grid resolution remains consistent with the global model. Future studies focused on India can use regional models with higher resolution to capture finer spatial details.

*Please revise Figure S6 to grid cell maps, not administrative regions, to provide more spatial details.*

**Response:** Thanks for your feedback. We have revised Figure S6 as follows:

[Figure]

Figure S6. The spatial distribution of CO from (a) ANTHRO emissions, (b) BB emissions, and (c) the ratio of CO from BB emissions to CO from ANTHRO emissions in 1995 and 2014. The changes from 1995 to 2014 are also presented.

*Let me rephrase my Comment 18: What uncertainties does the 95% CI in Figure 6 account for?*

**Response:** Thanks for your feedback. The 95% CI is derived from the RR estimates of long-term exposure to $PM_{2.5}$ and $O_3$, as described in Section 2.4, which assess the correlation between long-term exposure to $PM_{2.5}$ and $O_3$ and the mortality burden from specific diseases. Two-sided

P-values based on the likelihood ratio statistic were calculated to assess the significance of these estimates (Turner et al., 2016). We have revised manuscript as follows:

Line 295: The shaded area indicates the range of 95% confidence interval account for RR estimates of long-term exposure to $PM_{2.5}$ and $O_3$ (gray indicates half of the range).

References:

Turner, M. C., Jerrett, M., Pope, C. A., Krewski, D., Gapstur, S. M., Diver, W. R., Beckerman, B. S., Marshall, J. D., Su, J., Crouse, D. L., and Burnett, R. T.: Long-Term Ozone Exposure and Mortality in a Large Prospective Study, Am J Respir Crit Care Med, 193, 1134–1142, https://doi.org/10.1164/rccm.201508-1633OC, 2016.

*"Instead of India which could potentially have a higher value." is not clear and grammatically wrong. Please revise.*

**Response:** Thanks for your feedback. After reviewing the relevant literature, we identified an error with the content. Here's the revised version:

Line 356: Meanwhile, in estimating the mortality burden, we apply the RR derived from a global study, rather than using values specific to India, which could potentially be lower (Brown et al., 2022). Thus, our estimations for the air pollution-related mortality burden could be overestimated.

References:

Brown, P. E., Izawa, Y., Balakrishnan, K., Fu, S. H., Chakma, J., Menon, G., Dikshit, R., Dhaliwal, R. S., Rodriguez, P. S., Huang, G., Begum, R., Hu, H., D'Souza, G., Guleria, R., and Jha, P.: Mortality Associated with Ambient $PM_{2.5}$ Exposure in India: Results from the Million Death Study, Environmental Health Perspectives, 130, 097004, https://doi.org/10.1289/EHP9538, 2022.

*What do you mean by the following? "Finally, another limitation in our experimental design was that we set global fixed emissions for ANTHRO and BB instead of in India only, ignoring the impact of intercontinental transportation."*

**Response:** Thanks for your feedback. The sentence refers to a limitation in the experimental design of the study. The FixAN or FixBB scenarios fix the global ANTHRO and BB emissions to 1995 levels, rather than focusing solely on India. Therefore, this study overlooks the impact of intercontinental transportation, such as how pollutant changes from other countries or regions might affect air quality in India.

*Again, please check and correct all grammatical errors. E.g. "The ANTHRO emissions includes eight sectors" ...*

**Response:** Thanks for your feedback. We have carefully reviewed manuscript to correct the grammatical errors. Now we revise it as following:

Line 91: The ANTHRO emissions include eight sectors: agriculture; energy; industrial; transportation; residential, commercial, other; solvents production and application; waste and international shipping (Hoesly et al., 2018).